



# Vulnerable top-of-permafrost ground ice indicated by remotely sensed late-season subsidence

Simon Zwieback[1] and Franz J. Meyer[1]

[1]Geophysical Institute, University of Alaska Fairbanks, Fairbanks, AK, USA

**Correspondence:** S. Zwieback (szwieback@alaska.edu)

**Abstract.** Ground ice is foundational to the integrity of Arctic ecosystems and infrastructure. However, we lack fine-scale ground ice maps across almost the entire Arctic, chiefly because ground ice cannot be observed directly from space. Focusing on northwestern Alaska, we assess the suitability of late-season subsidence from Sentinel-1 satellite observations as a direct indicator of vulnerable excess ground ice at the top of permafrost. The idea is that, towards the end of an exceptionally warm summer, the thaw front can penetrate materials that were previously perennially frozen, triggering increased subsidence if they are ice rich. For locations independently determined to be ice rich, the late-season subsidence in an exceptionally warm summer was 4–8 cm (5th–95th percentile), while it was lower for ice-poor areas (-1–2 cm). The distributions overlapped by 2%, demonstrating high sensitivity and specificity for identifying top-of-permafrost excess ground ice. The strengths of late-season subsidence include the ease of automation and its applicability to areas that lack conspicuous manifestations of ground ice, as often occurs on hillslopes. One limitation is that it is not sensitive to excess ground ice below the thaw front and thus the total ice content. Late-season subsidence can enhance the automated mapping of vulnerable permafrost ground ice, complementing existing (predominantly non-automated) approaches based on largely indirect associations with vegetation cover and periglacial landforms. Improved ground ice maps will prove indispensable for anticipating terrain instability in the Arctic and sustainably stewarding its ecosystems.

## 1 Introduction

Permafrost conditions are changing across the Arctic, as evidenced by widespread observations of ground warming and increasing terrain instability (Romanovsky et al., 2010; Jorgenson et al., 2015; Segal et al., 2016; Box et al., 2019). It is primarily the presence of excess ground ice in the permafrost that makes an area susceptible to instability (French and Shur, 2010; Kanevskiy et al., 2017). As excess ice melts and the meltwater drains, the ground will settle, slump or collapse (Morgenstern and Nixon, 1971; Kokelj and Jorgenson, 2013; Shiklomanov et al., 2013). Even though such thermokarst is an infrastructure hazard, we lack accurate fine-scale estimates of excess ground ice over most of the Arctic (Heginbottom, 2002; Melvin et al., 2017). This lack is a major limitation for sustainably planning in the Arctic and for accurately anticipating how ecosystems and the hydrologic cycle will change (Prowse et al., 2009).

The paucity of fine-scale ground ice maps is largely due to the fact that permafrost ground ice is not directly observable from satellites (Heginbottom, 2002). Current approaches for mapping ground ice have significant shortcomings. One approach,

**Late-season subsidence and top-of-permafrost ground ice**

ice content profiles $e(y)$        thaw season subsidence

**Figure 1.** Late-season subsidence is predicted to be closely related to the excess ice content at the top of permafrost in an exceptionally warm summer. When the thaw front penetrates the permafrost in the late season, the melt of excess ice in the permafrost, where present, will give rise to increased subsidence. The subsidence time series (positive sign: downward movement) are referenced to zero at the beginning of the late season (second week of August; shown on the right in dark grey).

palaeogeographic modelling of ground ice aggradation and degradation, is currently limited to coarse-scale assessments (Jorgenson et al., 2008; O'Neill et al., 2019). For localized maps, the standard approach is to upscale costly field observations and expert interpretations based on imperfect indirect associations with vegetation cover and surficial geology (Pollard and French, 1980; Heginbottom, 2002; Reger and Solie, 2008; Paul et al., 2020). This works well where near-surface perennial

ground ice can be reliably excluded (such as under active floodplains; Jorgenson et al. (1998); Reger and Solie (2008)), or where there are robust indicators of excess ground ice that can be recognized using (largely manual) image analysis. These include aggradational landforms such as palsas (Borge et al., 2017) and, more commonly, degradational features that include thermokarst lakes, thaw slumps and ice wedge pits (Dredge et al., 1999; Farquharson et al., 2016; Kokelj et al., 2017; Zhang et al., 2018). Two problems with this approach are the paucity of reliable ground ice indicators in many areas (Mackay, 1990;

Jorgenson et al., 2008; Reger and Solie, 2008), for instance on hillslopes, and that the mapping based on degradational features works best when it is already too late. What is needed is an indicator that can be used to map ground ice vulnerable to future degradation.

    Here, we study the suitability of late-season subsidence in an exceptionally warm summer as an indicator of excess ground ice at the top of permafrost. The idea is that, towards the end of a hot summer, the thaw front can penetrate materials that

were previously perennially frozen (Fig. 1). If these materials do not contain excess ground ice, we generally do not expect to observe elevated late-season subsidence. If they are rich in excess ground ice, the melt of this vulnerable ice is predicted to induce a characteristic late-season acceleration of subsidence (Harris et al., 2011). The excess ice in the upper permafrost frequently occurs in the form of segregated ice as part of an ice-rich intermediate layer (Shur et al., 2005), as well as within ice





wedges that may or may not be protected by an overlying transient and intermediate layer (Jorgenson et al., 2015; Kanevskiy et al., 2017).

Late-season subsidence is a physically based indicator of vulnerable top-of-permafrost excess ground ice. To sketch the physical connection, we will make the simplifying assumption that even on submonthly time scales, thaw consolidation equals the melt of excess ground ice in a given period (Morgenstern and Nixon, 1971). The subsidence $s(t_1, t_2)$ between times $t_1$ and $t_2$ is then equal to the total excess ice that melts during that period:

$$s(t_1, t_2) = \int_{y_f(t_1)}^{y_f(t_2)} e(y)\, \mathrm{d}y, \tag{1}$$

where $e(y)$ is the excess ice content $[-]$ per unit depth $\mathrm{d}y$. $y_f(t)$ is the depth of the thaw front relative to the surface at the beginning of the thaw season; it is assumed to be a monotonic function of $t$. By judicious choice of $t_1$ and $t_2$, one can determine at which depth to probe the excess ice content. If the time period includes the early thaw season (small $y_f$), $s$ will reflect the seasonal ground ice (Lewkowicz, 1992; Chen et al., 2020). By focusing on the late season instead (larger $y_f$), we intend to isolate the excess ice at the base of the active layer and top of the permafrost (Harris et al., 2011; Bartsch et al., 2019). Because the observational strategy relies on ice melt, the identification of excess ice in the upper permafrost is tantamount to its vulnerability.

We assess the sensitivity and specificity of remotely sensed late-season subsidence as a permafrost ground ice indicator. To estimate late-season subsidence on regional scales, we use Sentinel-1 satellite radar interferometric observations (Bartsch et al., 2019; Wang et al., 2020). We test its suitability as a permafrost ground ice indicator in northwestern Alaska, contrasting the exceptionally warm summer of 2019 with preceding years. We do so by comparing it to ground ice cores and to an independent ground ice map, which we derived based on manual interpretation of high-resolution optical images and field observations. Because the manual interpretation relied on conspicuous indicators of ground ice, it was not applicable to featureless hillslopes – in contrast to late-season subsidence. Based on these assessments, we appraise the suitability and discuss the limitations of late-season subsidence as a ground ice indicator. These findings will serve to enhance the automated mapping of ground ice and anticipating terrain instability on pan-Arctic scales.

## 2 Study area

Our study area is located in the northwestern Alaskan Arctic, near the town of Kivalina (Fig. 2a). The surficial geology and topography of the study area is varied (DOWL Engineers, 1994; Tryck Nyman Hayes, 2006). The spectrum includes marine deposits near the mouth of the Kivalina river; various types of alluvial and colluvial sediments; as well as bedrock outcrops and rubble-covered surfaces at higher elevations. The area is underlain by warm (quasi-)continuous permafrost (Tryck Nyman Hayes, 2006).

Excess ground ice at the top of permafrost underlies many locations, as indicated by geotechnical investigations as well as remote sensing and field observations. Geotechnical analyses have been necessitated by environmental hazards such as

### Study area and meteorological conditions

a) Location and topography    b) Summer thawing degree days

**Figure 2.** The Kivalina study area in northwestern Alaska a) comprises areas of low to moderate topography (source: TanDEM-X DLR (2020)). b) Thawing degree days (TDD) estimated from MERRA-2 air temperatures identify 2019 as an exceptionally warm summer.

increasing storm surges and coastal erosion, which have been driving efforts to relocate the village from its present location on a low-lying barrier island. The 2006 master plan for the relocation planning project (Tryck Nyman Hayes, 2006) concluded that all investigated alternative sites were at least partially underlain by ice-rich permafrost. Ice wedges are abundant in old alluvial and lacustrine deposits (Shannon & Wilson, Inc., 2006). Ice-rich layers of segregated ice are also ubiquitous in fine-grained sediments (Shannon & Wilson, Inc., 2006).

Air temperatures in the summers of 2017 to 2019 differed markedly. 2019 was a record summer in western Alaska. According to MERRA-2 reanalysis data (Gelaro et al., 2017; Global Modeling and Assimilation Office, 2020) shown in Fig. 2b), the thawing degree days in 2019 exceeded those of the, with respect to the previous decade, average summer of 2017 by more than a third. Average daily temperatures in 2019 were particularly elevated in late June and early July (Fig. S1), and they also remained consistently above 10°C in late August and the first half of September. The summer of 2018 was also warm, but not as exceptional as that of 2019.

## 3 Methods

### 3.1 Subsidence from radar interferometry

#### 3.1.1 Sentinel-1 observations

To estimate surface displacements at a resolution of 60 m, we used Sentinel-1 observations between early June and mid-September 2017–2019. The Sentinel-1 observations (Torres et al., 2012), acquired at a frequency of 5 GHz with VV polarization in the Interferometric Wide mode, were available at 12-day intervals from an easterly look direction (37 degrees incidence angle; path 15, frame 367). There was one exception in 2017, during which a gap of 18 days occurred.





### 3.1.2 Estimating subseasonal subsidence time series

To estimate a subseasonal displacement $d$ time series from the Sentinel-1 observations Copernicus Sentinel (2020), we ap-
plied Short BAseline Subset (SBAS) processing (Berardino et al., 2002). The rationale of this Interferometric SAR (InSAR)
technique is to derive displacement time series from redundant interferograms. We first formed interferograms with temporal
separations of up to 24 days, using spectral diversity techniques for coregistration (Scheiber and Moreira, 2000) and removing
the topographic phase contribution using the TanDEM-X DEM (Rizzoli et al., 2017; TanDEM-X DLR, 2020). After multi-
looking, we unwrapped the interferograms using SNAPHU (Statistical-Cost, Network-Flow Algorithm for Phase Unwrapping;
Chen and Zebker (2001)). Ionospheric phase corrections were deemed to be unnecessary. We then estimated the displacement
time series, with a temporal sampling of 12 days, from the interferogram stack. We used a weighted least squares approach
based on the singular value decomposition (Berardino et al., 2002), with the weights determined by the Cram'er-Rao phase
variance estimate (Tough et al., 1995).

The time series are reported as displacements $d$ along the line-of-sight direction, with positive values corresponding to
increasing distance. Owing to the hilly terrain, we chose not to convert them to vertical displacements. However, we did
not discern aspect-dependent trends that are associated with downslope movements. For simplicity's sake, we will refer to
displacements with increasing distance as subsidence. To emphasize the late-season subsidence, we set the zero point of the
relative subsidence time series to be at the beginning of the late season $t_1$.

### 3.1.3 Referencing and assessing its quality

To reduce long-wavelength atmospheric errors in the subsidence observations, we spatially referenced the raw time series at
multiple locations with outcropping bedrock or a thin rubble veneer. These locations were assumed to be stable (Reger and
Solie, 2008; Antonova et al., 2018).

To assess the quality of the referencing, we chose eight bedrock validation points distributed across the study region. Assum-
ing these points to remain stable, the subsidence observations at these locations are a measure of the observational uncertainty
(Zwieback et al., 2018). This uncertainty estimate subsumes decorrelation errors and uncompensated atmospheric contributions
under a single number.

### 3.1.4 Late-season subsidence from spline fitting

We estimated the late-season subsidence $d_l$ by fitting spline basis functions to the referenced subsidence time series. The
advantage of fitting a flexible and yet simple spline function is that measurement noise, such as residual atmospheric errors,
can be reduced (Berardino et al., 2002). To capture a range of subseasonal subsidence patterns, we used three cardinal quadratic
B-splines for the subsidence rate (first derivative), corresponding to the cubic spline basis functions for the subsidence time
series shown in Fig. 3a).

**Late-season subsidence from spline fitting**

**Figure 3.** a) The three cubic spline basis functions used for estimating early–mid-season and $d_e$ late-season $d_l$ subsidence. b) Estimated $d_e$ and $d_l$ over the validation points, with the individual values for all points and years shown at the top, and a kernel density estimate of the distribution below.

The late-season subsidence $d_l$ was defined to be the cumulative subsidence between 10 August ($t_1$) and 10 September ($t_2$) for all years, the end point having been chosen to minimize the confounding impact of diurnal frost heave (Chen et al., 2020). We derived $d_l$ from the spline fit.

We also estimated total subsidence during the early and mid thaw season (10 June to 10 August), $d_e$. While $d_e$ is not expected to contain direct information about permafrost ground ice, being sensitive to soil texture and interannual variations in active layer moisture (Lewkowicz, 1992; Harris et al., 2011), it provides a simple reference for assessing the late-season speed-up characteristic of melting top-of-permafrost excess ground ice.

## 3.2 Assessment with independent ground ice data

### 3.2.1 Manual mapping

We assessed the suitability of late-season subsidence as an indicator of permafrost ground ice content by comparing the satellite observations to an independently derived ground ice map. The map comprised two primary categories: ice rich and ice poor. The sensitivity and specificity of late-season subsidence were assessed from the observed distributions conditional on the location being ice rich and ice poor, respectively.

The independently derived ground ice map was obtained using manual interpretation and expert knowledge, drawing on field observations and high-resolution (∼1 m) satellite imagery. The mapped focus area, 8 km by 8 km in size, was chosen because of the wide range of ecotypes and the availability of field observations. A drawback of the map are its gaps, as ice-rich permafrost cannot be identified reliably in areas that lack unambiguous indicators. Furthermore, six percent of the area were discarded in the comparison to avoid unrepresentative values over lakes and infrastructure.

The key considerations in the manual mapping were:

1. Ice-rich permafrost was assigned to ecotypes where high-resolution imagery and field observations revealed direct indicators of excess ice at the top of permafrost. Ice wedge polygons – some only barely visible or inconspicuous except





along lakeshores and beaded streams, others in an advanced state of degradation – are widespread (Fig. 4). They are

abundant in old alluvial and lacustrine deposits (DOWL Engineers, 1994; Shannon & Wilson, Inc., 2006), as well as in colluvial sediments that are rich in retransported silt (DOWL Engineers, 1994). Pingos in drained lake basins are also direct indicators of excess ground ice (Mackay, 1973).

2. Exposed bedrock and rubble-covered surfaces (Fig. 4b) were classified as ice poor. This classification is generally supported by geotechnical investigations and field evidence from the study area (Pewe et al., 1958; DOWL Engineers, 1994).

Deviations from this general pattern cannot be excluded (Robinson and Pollard, 1998; French and Shur, 2010).

3. Active and recent inactive floodplains were categorized as ice poor (Fig. 4a; DOWL Engineers (1994)) because near-surface permafrost, where present, is too young for abundant ground ice to have aggraded (Jorgenson et al., 1998). We did not observe any indications of abundant excess ground ice in these ecotypes.

4. Such indirect indicators as vegetation cover and surficial geology were used to classify areas as ice rich where similar

adjacent areas were clearly ice rich. For instance, if polygons were visible over 80% of an alluvial deposit, the entire deposit was classified as ice rich (Fig. 4a).

5. The ice content of the remaining locations was deemed to be indeterminate. The majority of these locations are in uplands and on hillslopes, where ice content is inherently variable (Morse et al., 2009). In Fig. 4b, the areas above the clearly ice-rich toe and midslope deposits (Fig. 4b) lack visible manifestations of ice-rich permafrost, but it is known that some

are ice rich (Shannon & Wilson, Inc., 2006). Floodplain deposits of intermediate age (between recent inactive and old abandoned floodplains) were also classified as indeterminate (Jorgenson et al., 1998).

The mapping process is inherently subjective. The discretization of top-of-permafrost excess ground ice into a limited number of categories present challenges (Tryck Nyman Hayes, 2006; Paul et al., 2020). The most difficult decisions were about where to draw the line between the ice-rich/ice-poor and the indeterminate category.

### 3.2.2   Cores

We also compared the late-season subsidence observations to geotechnical assessments of permafrost ground ice content from >3 m deep cores. The cores were taken in 2005 by Shannon $ Wilson, Inc. to investigate the suitability of three proposed relocation sites for Kivalina (Shannon & Wilson, Inc., 2006). Ice-rich permafrost was encountered underneath all sites, but the Tatchim Isua site also encompassed an ice-poor bench. The ice-rich nature of the area surrounding the ice-poor bench was not

evident at the surface; it only became conspicuous at the surface ~300 m upslope, in the form of faint polygons (Shannon & Wilson, Inc., 2006).

Although coring is the most reliable method for determining ground ice content, the spatial and temporal representativeness of the cores need to be considered. The point observation a single core represents could paint a misleading picture of ground ice content (Morse et al., 2009; Kanevskiy et al., 2017) when compared to a ~60 m resolution cell. The cores were further

taken a decade before the remote sensing observations. However, none of the locations showed signs of severe disturbance, and





**Manual mapping of permafrost ground ice**

a) Alluvial & floodplain

b) Colluvial & bedrock

**Figure 4.** Manual classification into ice-rich and ice-poor terrain relies heavily on visible manifestations of ground ice. a) Ice-rich predominantly alluvial deposits on the left, as indicated by ice wedge polygons (inundated troughs; heterogeneous vegetation communities; cut bank morphology), whereas the active floodplain of the Wulik river on the right is ice poor. b) Catena with the ice-poor nature of the ridge (right) being indicated by bedrock and rubble; there is little evidence for or against ice richness near the top of the slope (middle); the midslope hosts abundant but faint polygons (inset). For visual display purposes, the boundaries have been shifted.

the observed thickness of the ice-rich layer at the top of permafrost, where present, exceeded 30 cm (Shannon & Wilson, Inc., 2006).

We classified the cores as ice poor and ice rich based on the descriptions in the geotechnical report (Shannon & Wilson, Inc., 2006). The ground ice content of each core was summarized verbally as ice rich or ice poor, complemented by pictures and estimates of the visible ice content. All cores but one were ice rich with visible ice contents >30% in the form of segregated and massive ice. The only ice-poor core was a gravelly soil grading into weathered bedrock (5% visible ice content in joints). Two cores taken from within the same ice wedge polygon were combined for the comparison exercise. Tab. S1 summarizes the locations and ground ice properties of all 13 coring locations.

## 4 Results

### 4.1 Spatial and temporal variability of late-season subsidence

#### 4.1.1 Regional variability

The observed late-season subsidence in our study area showed two distinct peaks in 2019 (Fig. 5a). One corresponded to no or very small subsidence (-1 – 1 cm). The other corresponded to regions with elevated subsidence (∼5 cm) in 2019. In 2017, the subsidence in these regions was lower (1–4 cm). There were no notable instances of pronounced negative late-season subsidence.



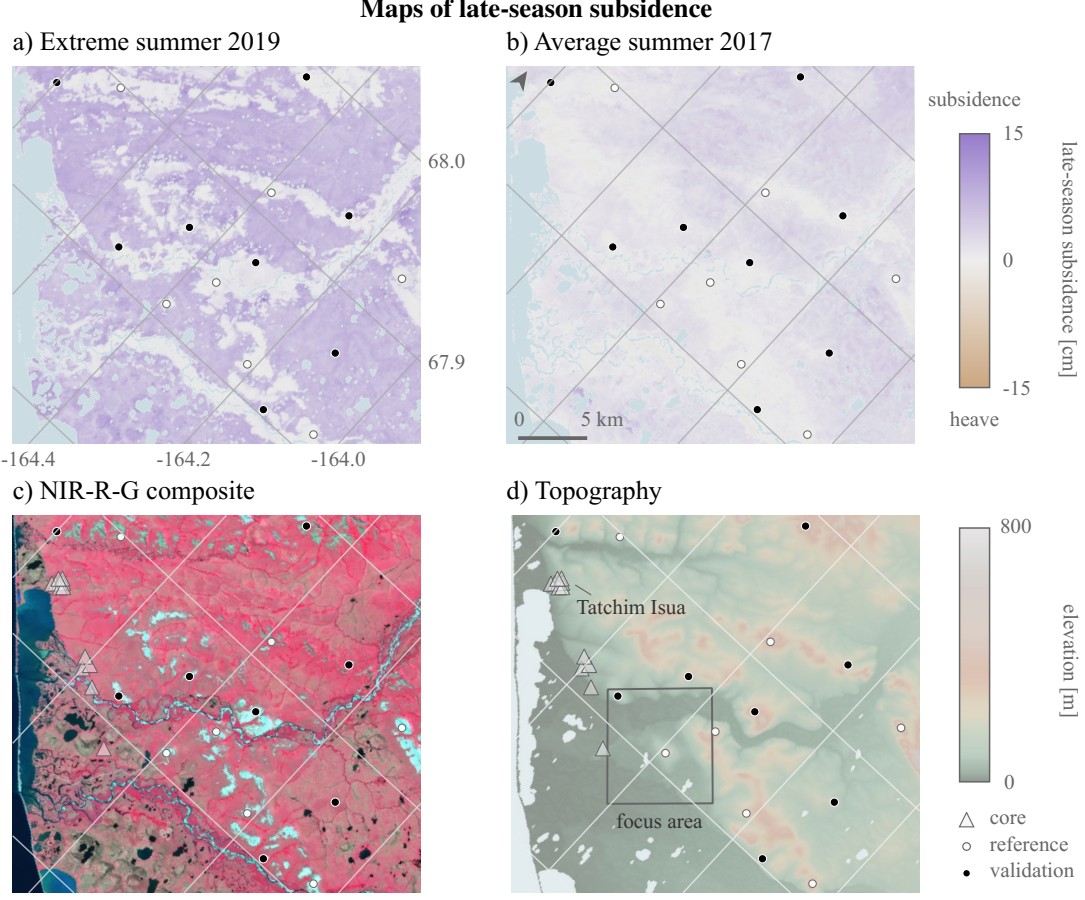

**Figure 5.** Regional variability of remotely sensed late-season subsidence $d_l$ within the study area. a) $d_l$ in the exceptionally warm summer of 2019; b) $d_l$ in the average summer of 2017; c) Sentinel-2 false-colour composite image (Copernicus Sentinel, 2020); d) Topography estimated from the TanDEM-X DEM. The reference and validation points for the Sentinel-1 subsidence estimates are indicated by white and black circles, respectively; the locations of the ground ice cores are shown by triangles. The focus area for manual mapping and the Tatchim Isua candidate relocation site are shown in d).





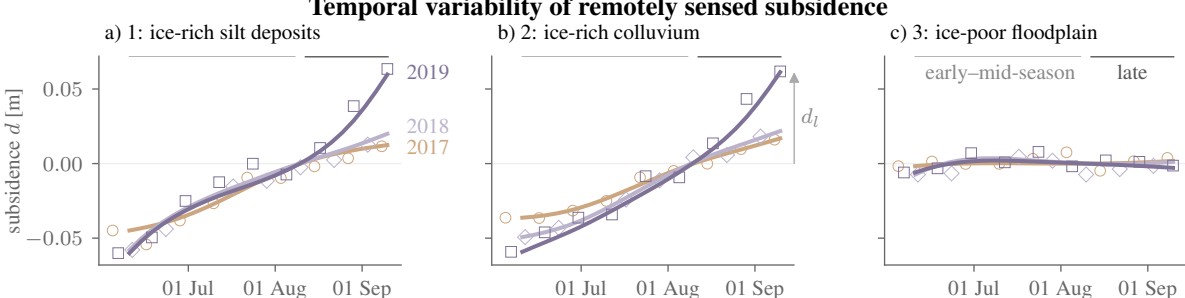

**Figure 6.** Subsidence time series (line: spline fit, markers: unconstrained) from all three years. The locations shown in a–c) are shown in Fig. 7d.

Late-season subsidence was consistently low in elevated uplands (e.g. bedrock or talus without vegetation cover) and along rivers (recent floodplains without or with dense vegetation cover; Fig. 5a–b). This association supports the assumption that the reference and validation points on bedrock (circles in Fig. 5) by circles, could be assumed to be stable. The early–mid and late-season subsidence observations at the validation points, shown in Fig. 3b) had a root-mean-square deviation of 0.6 cm.

Other low-lying areas and many hillslopes exhibited elevated late-season subsidence, especially in the exceptionally warm summer of 2019 (Fig. 5). Late-season subsidence commonly exceeded 5–8 cm in the exceptionally warm summer of 2019, whereas it rarely exceeded 3–5 cm in the average summer of 2017.

### 4.1.2   Temporal variability

The inter-annual variability of the observed subsidence time series is illustrated by three examples in Fig. 6. The late-season
subsidence in 2019 exceeded 5 cm in the ice-rich (ice wedges) deposits shown in panels a and b, respectively. According to the radar observations, there was an acceleration in the rate of subsidence in the late thaw season, as the rate more than doubled. The rate and the total subsidence during the late season is approximately a factor of three smaller in the years 2017 and 2018. For the floodplain in Fig. 6c, the interannual variability was less than 1 cm. The late-season subsidence was of a magnitude comparable to the observational uncertainty in all years at this ice-poor location.

### 4.2   Assessing the suitability as an indicator of top-of-permafrost ground ice

### 4.2.1   Suitability in an exceptionally warm summer

Late-season subsidence in the hot summer of 2019 was markedly different for ice-rich and ice-poor areas (Fig. 7a). It exceeded 4 cm at all cored locations rich in top-of-permafrost ground ice. For the ice-rich areas, as determined by independent manual mapping (Fig. 7c), $d_l$ varied between 4 and 8 cm (5th and 95th percentile; Fig. 7b. For for ice-poor areas the range was -1 to 2
cm. Only in the extreme tails (2%) do the distributions overlap, ensuring a robust separability of the two classes.





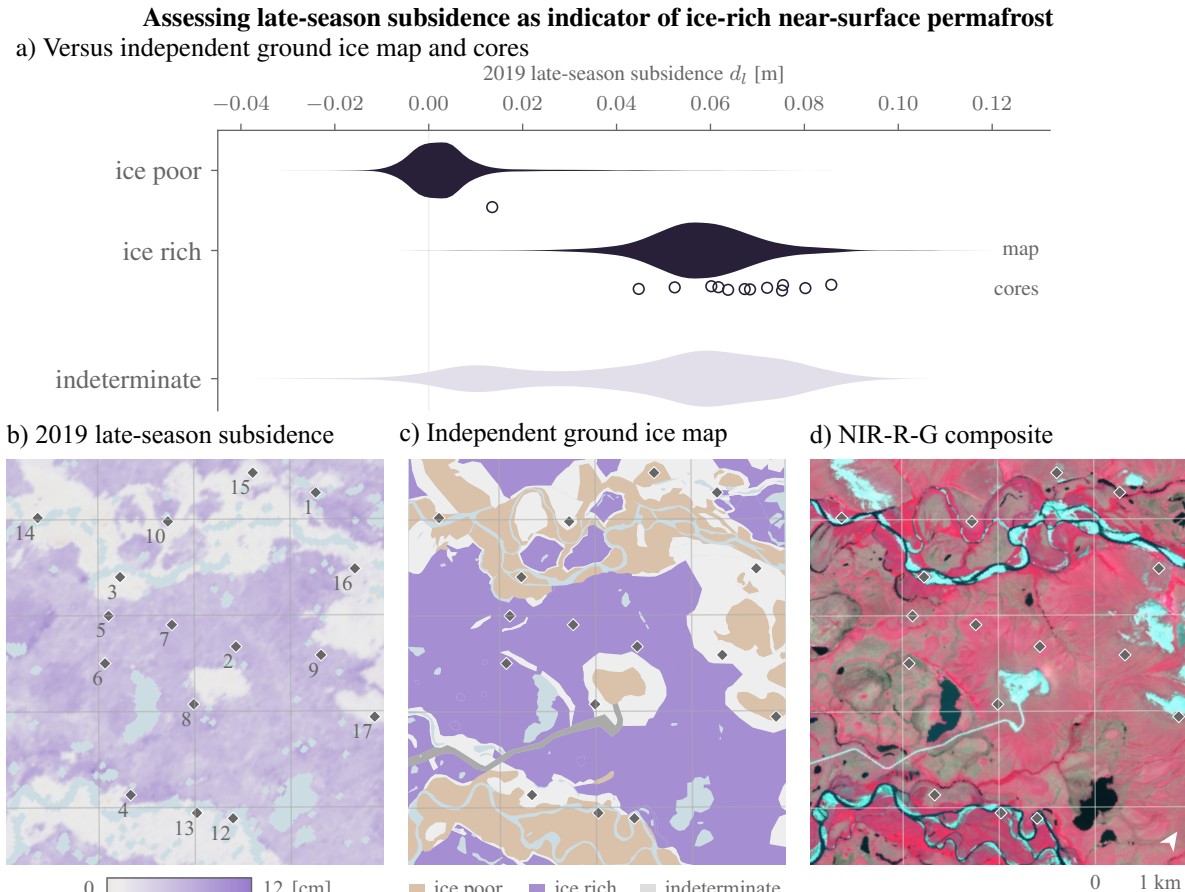

**Figure 7.** a) Little overlap between the distributions of late-season subsidence $d_l$ in 2019 over ice-rich and ice-poor areas, as independently determined by manual mapping in the focus area. The markers just below the kernel density estimates further show the observations at the boring locations (triangles in Fig. 5). b) The estimated $d_l$; c) the independently determined ground ice classification; and d) Sentinel-2 false-colour composite (Copernicus Sentinel, 2020) for the focus area defined in Fig. 2. The diamonds indicate points mentioned in the text.





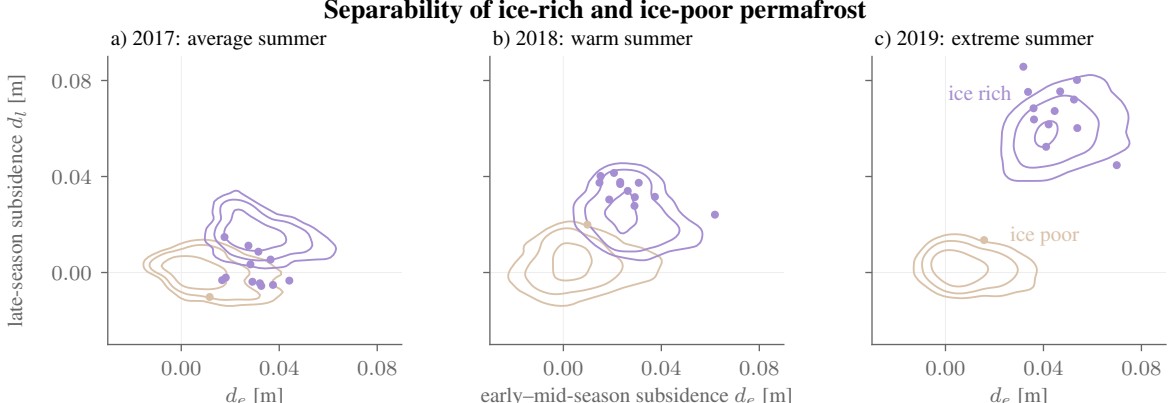

**Figure 8.** Contours plot of a kernel density estimate of the early–mid-season and late-season subsidence for ice rich (purple) and ice poor (orange), as determined independently by manual mapping (Fig. 7c). The markers correspond to the values observed at the location of the ice cores (triangles in Fig. 5)

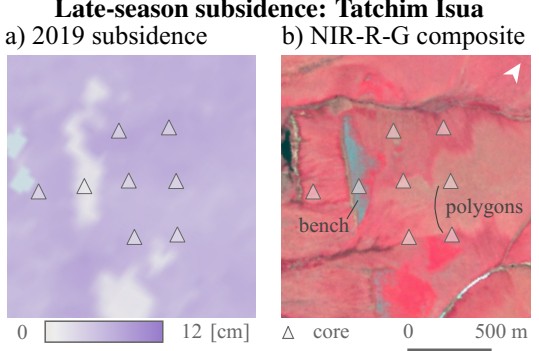

**Figure 9.** The late-season subsidence in 2019 a) at the Tatchim Isua site is smaller at the site of the gravelly bench, which appears grey in the false-colour composite (courtesy of Planet Labs, Inc.; Planet Team (2020)) in b), than the areas further upslope (right) and downslope (left). The triangles indicate the location of the ice cores (cf. Fig. 5): the one on the bench was ice poor, the others ice rich. Ice wedge polygons were observed in the field ∼300 m upslope from the bench.

The separability based on the 2019 late-season subsidence $d_l$ is better than that based on the early–mid-season subsidence $d_e$. The $d_e$ distributions of ice-rich and ice-poor areas overlap (Fig. 8c), whereas the $d_l$ distributions are concentrated around two separate peaks (see also Fig. 7a).

The candidate relocation site Tatchim Isua was characterized by a narrow zone with low late-season subsidence (Fig. 9).
This ∼100 m wide zone largely coincides with the gravel-covered bench that a single core from 2005 indicates to be ice poor (Shannon & Wilson, Inc., 2006). Late-season subsidence was elevated (∼7 cm) at the location of the seven cores further

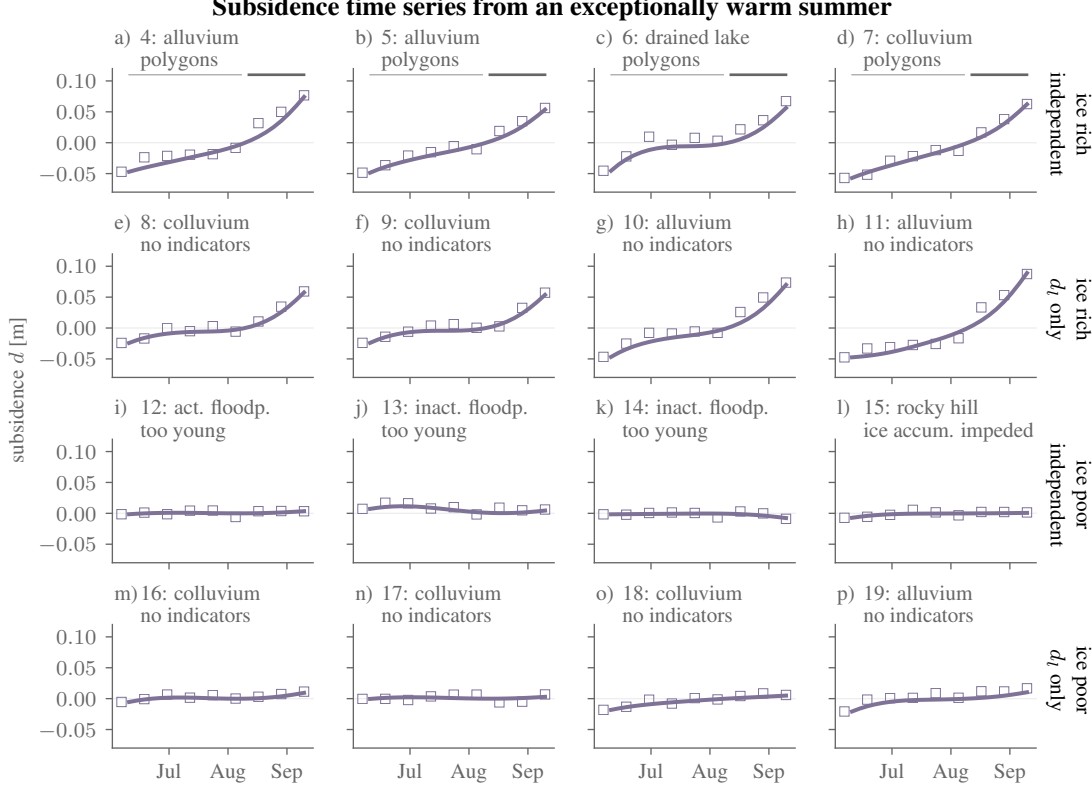

**Figure 10.** Subsidence time series (line: spline fit, markers: unconstrained) from 2019 for points from 7d). First row a–d): points that were independently determined to be ice rich; second row e–h) indeterminate according to manual mapping, but subsidence indicates they are ice rich; third row i–l): independently determined to be ice poor; fourth row m–p): indeterminate according to manual mapping, but subsidence indicates they are ice poor.

downslope or upslope. All cores contained ice-rich materials at the top of permafrost, but in the field the ice-rich nature was not readily apparent at the proximal coring locations (Shannon & Wilson, Inc., 2006). Visible manifestations of ground ice, in the form of faint polygons, were observed ∼400 m upslope from the bench.

Further examples from a range of geologic settings serve to illustrate the suitability of $d_l$ for identifying ice-rich permafrost. Fig. 10a–d shows instances of ice-rich permafrost with ice wedge polygons. They all exhibited elevated $d_l$ of 4–8 cm, corresponding to an increased subsidence rate during the late season. Conversely, the observed subsidence was below 1 cm for the points shown in Fig. 10i–l, which were independently determined to be poor in ground ice because of their young age, such as active and inactive floodplains, or because the formation of excess ice is impeded by their composition.

The most interesting cases are those areas where the manual ground ice mapping was indeterminate because there was no strong evidence for either category. They exhibited a bimodal distribution of $d_l$ (Fig. 7a). The larger mode $d_l \sim 5$ cm, comparable to those of ice-rich areas. Examples include colluvial hillslope deposits without conspicuous ground ice indicators





(Fig. 10e–h). The characteristic late-season acceleration in subsidence leading to elevated $d_l \sim 5$ cm uniquely indicate that the top of the permafrost is ice rich. The smaller mode $d_l \sim 1$ cm roughly matches the observations over ice-poor terrain. The negligible subsidence observed at the locations shown in Fig. 10m–p suggests that the materials that thawed late in summer contained little excess ice.

### 4.2.2   Suitability in cooler years

In the average year of 2017, the suitability was reduced because the late-season subsidence distributions of ice-rich and ice-poor regions overlapped substantially (Fig. 8a). The late-season subsidence $d_l$ of 80% of the terrain that was mapped to be ice rich and all the coring locations was less than 2 cm. On average, it was a factor of 5 smaller than during 2019.

In the warm summer of 2018, the separability based on the $d_l$ distributions was better than in 2017. The distributions overlapped at the 10% level (Fig. 8b), compared to 2% in 2019. This is largely due to the smaller late-season subsidence of ice-rich terrain compared to the exceptionally warm summer of 2019.

## 5   Discussion

### 5.1   Suitability for identifying vulnerable top-of-permafrost ground ice

Comparing the late-season subsidence to the independently determined ground ice map and the ice cores, we note that ice-poor and ice-rich permafrost are well distinguishable in the exceptionally warm summer of 2019 (Fig. 7). Ice-poor areas were stable, whereas ice-rich areas exhibited pronounced ($\sim 5$ cm) late-season subsidence 10. We suggest that the elevated late-season subsidence was caused by the melt of vulnerable top-of-permafrost ground ice.

The late-season speed-up in subsidence is interpreted as the thaw front penetrating ice-rich materials at the top of permafrost (Harris et al., 2011). A characteristic feature is that the subsidence rate increased up to fivefold in late summer (Fig. 10). This acceleration is thought to stem from the contrast between the ice-rich intermediate layer (or massive ice) and the comparatively ice-poor lower half of the active layer and the transient layer, where present (Matsuoka, 2001; Shur et al., 2005; French and Shur, 2010). In practical terms, the acceleration increases the robustness of the separability with respect to the chosen starting day of the late season period, thus facilitating the identification of vulnerable top-of-permafrost ground ice.

Elevated late-season subsidence indicates that top-of-permafrost excess ice is present and vulnerable. Its is vulnerable because subsidence corresponds to initial degradation, under the assumptions of Eq. 1. If the ice-rich layer is thick, late-season subsidence can be a subtle precursor for long-term terrain instability (Kanevskiy et al., 2017).

Inter-annual variability in late-season subsidence of ice-rich areas poses challenges for ground ice mapping. Potential sources of inter-annual variability and trends include surface changes (e.g., soil moisture, disturbance), complete thaw of the ice-rich layer, and the meteorological conditions (Shiklomanov et al., 2010; Bartsch et al., 2019). Figure 6 shows ice-rich areas with only limited ($\sim 2$ cm) late-season subsidence in the warm summer of 2018, which apparently was not warm enough to allow





for reliable identification of the vulnerable ground ice at this location. That the identification strategy presupposes an initial degradation of ground ice constitutes its biggest limitation.

## 5.2 Limitations

The excellent separability in our study area in the exceptionally warm summer does not imply that interferometric observations of late-season subsidence are a universally applicable basis for mapping ice-rich permafrost. Limitations arise from observational uncertainties and from the imperfect sensitivity and specificity of subseasonal subsidence as an indicator of top-of-permafrost excess ground ice.

### 5.2.1 Observational uncertainties

Observational uncertainties chiefly arose from errors in the referencing, dominated by uncompensated atmospheric contributions, and from location-specific systematic and random errors.

The errors due to imperfect referencing were determined to be substantially smaller (1 cm; 3b) than the typical difference in late-season subsidence between ice-rich and ice-poor permafrost terrain (5 cm; Fig. 7a).

This uncertainty metric does not account for systematic biases associated with changes in soil or vegetation moisture, as it was obtained from rocky, non-vegetated surfaces. Soil moisture generally increases toward the end of the thaw season, which would correspond to a spurious subsidence signal (De Zan et al., 2015). However, the worst-case estimates of the bias ($\sim$ 1 cm at C-band) are a factor of five smaller than the late-season subsidence observations (Zwieback et al., 2017). Vegetation moisture in shrubs will decrease with senescence, corresponding to a spurious heave signal (Zwieback and Hajnsek, 2016). The persistently small magnitude of the displacement estimates over shrub-covered inactive floodplains (Fig. 5) indicates that these systematic errors were not a major confounding factor in the present study. However, dedicated in-situ observations are needed to accurately characterize the observational uncertainties. We expect these errors to be greater in densely vegetated, often discontinuous, permafrost (Wang et al., 2020).

### 5.2.2 Limitations of late-season subsidence for identifying top-of-permafrost ground ice

Late-season subsidence in a warm summer is not a perfect indicator of whether the top of the permafrost is ice rich or not.

Its sensitivity is impaired when excess ground ice does not manifest as elevated late-season subsidence. Such false negatives were widespread in the warm summer 2018 (Fig. 8b), indicating that the thaw front did not penetrate substantially into the ice rich materials in the upper permafrost (Harris et al., 2011). The ice content at the very top of permafrost can be reduced because the transient layer is subject to thawing in occasional warm summers (Shur et al., 2005). It then takes an exceptionally warm summer such as 2019, or disturbances like vegetation die-off (Jorgenson et al., 2015), for deep thaw of the intermediate layer or massive ice bodies to enhance late-season subsidence.

False negatives – even in an extremely warm summer – are expected to occur most commonly in the discontinuous and sporadic permafrost zone. There, disturbances such as forest fires are more likely to have obliterated the ecosystem-protected





or ecosystem-driven perennial ground ice near the surface (Jorgenson et al., 2010; Kanevskiy et al., 2012, 2014; Paul et al.,

2020), but see Burn (1997). Ice-rich permafrost may occur at depth, perhaps under a thick talik. The thermal regime of such permafrost is more complex (Jorgenson et al., 2010; Connon et al., 2018), and melt-induced subsidence does not necessarily occur primarily at the end of the thaw period. The sensitivity may further be diminished by sinkhole formation and piping underneath cohesive materials such as peat (Osterkamp et al., 2000); or by the retardation of thaw consolidation due to inefficient drainage of the excess melt water (Morgenstern and Nixon, 1971).

The specificity is impaired when unrelated processes induce late-season subsidence. Gradual subsidence due to processes such as organic layer degradation (Stephens et al., 1984) may be distinguished from the ground ice signals shown in Fig. 6a–b by the late-season acceleration typical for ice-rich permafrost. Elevated late-season subsidence may also reflect ice content at the bottom of the active layer. Cold permafrost and ample moisture supply promote ice segregation at the base of the active layer, which can be continuous with the intermediate layer (Mackay, 1981). Deformation related to precipitation events and

lateral flow may pose additional challenges, in particular in peatlands and on slopes (Roulet, 1991; Matsuoka, 2001; Gruber, 2020; Zhang et al., 2020). In hilly terrain it will be advantageous to resolve the downslope and surface-normal movement components, but we currently lack adequate satellite observations to do so routinely. The limited downslope movements in our study area likely contributed to the high specificity of late-season subsidence for mapping ice-rich permafrost.

A final limitation of this study and the preceding discussion is that the complexity of ground ice was simplified to just two

categories: ice rich and ice poor. In reality, however, excess ground ice content is a continuous parameter (Morse et al., 2009; Kanevskiy et al., 2012; Paul et al., 2020) whose magnitude could be constrained using late-season subsidence observations. A quantitative assessment at the appropriate observational scale will require densely sampled ground ice cores (Morse et al., 2009; Jorgenson et al., 2010).

### 5.3   Enhancing automated ground ice mapping

Late-season subsidence can enhance the automated mapping of vulnerable permafrost ground ice. Remotely sensed late-season subsidence can be mapped on pan-Arctic scales, thanks to the global availability of Sentinel-1 data. The mapping can be automated, as no manual interpretation an no calibration using in-situ cores are required.

Late-season subsidence is complementary to state-of-the-art mapping approaches based on visible manifestations of ground ice and indirect associations with vegetation cover or topographic variables. The greatest contribution of late-season subsidence

will likely be where field observations are sparse or where ground ice is indistinct and poorly correlated with surface characteristics. Examples includes uplands and hillslopes (Fig. 9), as well as areas underlain by ice wedges from the early Holocene or the Pleistocene (Pewe et al., 1958; Dredge et al., 1999; Reger and Solie, 2008; Morse et al., 2009; Jorgenson et al., 2015; Farquharson et al., 2016).

Incorporating geological constraints can counteract weaknesses of remotely sensed late-season subsidence. These include

larger errors in vegetated areas (Wang et al., 2020) or false negatives when the thaw front fails to penetrate deep into the ice-rich materials (Shur et al., 2005). Most importantly, geological and other observational constraints will be needed to estimate total excess ice contents.





# 6 Conclusions

We studied the late-season subsidence of permafrost terrain in northwestern Alaska. We predicted that ice-rich near-surface
permafrost would become detectable by enhanced subsidence toward the end of an exceptionally warm summer. By comparing
Sentinel-1 satellite observations of subsidence with independent ground ice data, we assessed the suitability of late-season
subsidence as an indicator of excess ground ice at the top of permafrost. Our principal findings and conclusions are:

1. In the exceptionally warm summer of 2019, the late-season subsidence observations were large ($\sim$ 4–8 cm) in areas that
   were independently determined to be rich in top-of-permafrost ground ice. The acceleration of subsidence is consistent
   with the degradation of vulnerable excess ice in the upper permafrost. Conversely, the observed late-season subsidence
   was small (-1–2 cm) in ice poor areas.

2. Distinguishing ice-rich from ice-poor terrain worked best in the exceptionally warm summer, as the respective late-
   season subsidence distribution overlapped by less than 2%. In the preceding summers the overlap was 10% or larger.

3. Late-season subsidence can enhance the mapping of vulnerable excess ground ice and the susceptibility to terrain insta-
   bility. A major strength is that it does not require conspicuous manifestations of excess ground ice, including degrada-
   tional landforms. Late-season subsidence in a warm summer may be relatively small, but it can serve as an observable
   precursor for much larger terrain changes.

4. The suitability of late-season subsidence as an indicator of top-of-permafrost excess ice will not always be as high as in
   this study. The greatest drawback is the lack of sensitivity when the thaw front does not penetrate deep into the ice-rich
   layers.

5. Remotely sensed late-season subsidence complements established techniques for estimating ground ice contents, such
   as manual mapping approaches that exploit visible manifestations of ground ice and indirect associations with surface
   characteristics. Because late-season subsidence is insensitive to excess ice at depth, it will be essential to incorporate
   geological reasoning and indirect associations established using in-situ observations for estimating total ice contents.

6. Its practical advantages for mapping ice-rich permafrost include the pan-Arctic availability of data, the ease of automa-
   tion, and the independence from costly in-situ observations.

Pan-Arctic expansion of the ground ice mapping using late-season subsidence is timely and societally relevant. It is timely
because of the widespread warming and accelerating degradation of permafrost. It is relevant because we lack accurate, fine-
scale ground ice maps over essentially the entire Arctic. Late-season subsidence observations can make a vital contribution to
anticipating terrain instability in the Arctic and to sustainably stewarding its vulnerable ecosystems.

*Data availability.* The subsidence estimates and the independent ground ice data have been published as Zwieback (2020b) and Zwieback
(2020a), respectively. The Sentinel-1 and 2 data are freely available from Copernicus Sentinel (2020), and the MERRA-2 data from Global



Modeling and Assimilation Office (2020). The Planet and TanDEM-X DEM data are available from Planet Team (2020) and TanDEM-X DLR (2020), respectively.

*Author contributions.*  SZ conceived of the idea and analysed the data. FJM provided guidance on the study design and the data processing. Both authors contributed to the writing of the manuscript.

*Competing interests.*  The authors declare no conflict of interest.

*Acknowledgements.*  The authors thank Trent Hubbard and Gabriel Wolken from the Alaska Division of Geological & Geophysical Surveys for sharing field notes and their knowledge about the Kivalina area. They are grateful to Vladimir Romanovsky for discussions about thaw
consolidation.

The authors acknowledge support by the National Aeronautics and Space Administration, grant number 80NSSC19K1494.



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
