# Peer review of "Vulnerable top-of-permafrost ground ice indicated by remotely sensed late-season subsidence"

_The Cryosphere, 2020_

## Referee Comment (RC1) · Anonymous Referee #1 · 2 Dec 2020

Zwieback and Meyer studied the suitability of exploiting InSAR-based late-season subsidence measurements in northwestern Alaska as an indicator of excess ground ice at the top of permafrost. They present a great piece of work that shows the value of radar remote sensing to upscale ground ice mapping and identify potentially unstable terrain at large scale. This topic is relevant both for Arctic research and operational end-user exploitation. The paper is well suitable for a publication in The Cryosphere and likely to become a reference in remote sensing of permafrost. I have no major concern regarding the methods and main results, but I think the manuscript could benefit from some additional information, clarifications, and adjustments, especially in figures. These elements, so called 'main comments', are described thereafter. I also listed

more technical suggestions, so called 'complementary comments', in the second part of the review.

———————

Main comments:

———————

Abstract:

- Some parts of the current version are not easy to understand without having read the whole paper (although this should be the point of an abstract). For ex l.6 'For locations independently determined to be ice rich': Maybe sth like 'Compared to an independently-generated manual mapping of ice-rich and ice-poor areas, . . .'; l.7-8 'The distributions overlapped by 2%...': the first time I read it, I understood 'spatial distribution' which obviously did not make sense. Maybe sth like: 'the distributions of the late-season subsidence values in referenced ice-rich/ice-poor areas overlap by only 2%, which. . .'.

- Seems that it was a conscious decision all along the article not mentioning that the subsidence results are InSAR-based. I understand that the detection technique is not the main point here but still think it would be good to mention it a couple of times, at relevant locations. For instance, in the abstract (l.3) and conclusions (l.326).

- Somehow l.58-66 do a better job to clearly summarize the work. Could maybe be used to rework the abstract?

———

Figures: I believe some figures could become easier to read with minor changes. Here are some suggestions:

- Figure 1 right: A bit confusing that subsidence is written on the left (y-axis) but subsidence-heave is indicated on the right (in addition with counter-intuitive directions).
Indicate that the black horizontal line in the center corresponds to 0 and maybe consider inversing the direction (heave upwards, subsidence downwards). Except if there is really a good reason not doing so, it would be much easier to read. Positive sign mentioned in the legend is anyway not shown in the figure and I guess (+/-) here is used to correspond to changes of sensor-to-ground distance, but it should not impact the visualization (if you imagine Sentinel-1 at the top of the page and the ground at the location of your horizontal straight line, an increase of distance goes downwards). Consider also having all text in black instead of light grey.

- Figure 2 left: What do the different colors of the spline lines actually indicate? Missing a legend.

- Figure 4: Maybe add an information about location. In legend: what does Catena mean here? Is it a place?

- Figure 5: As you refer to 5-8 cm/yr of typical values, I wonder if your subsidence maps would not benefit from a better contrast by reducing min/max values of your color scale (-10 to 10?). Or considering other colors or an asymmetric scale (I guess you do not have heave values up to -15). Add in legend somewhere that blue = no data in a) / b), i.e. under a certain coherence threshold I guess (which btw is not mentioned). Remind somewhere that late-season is 10 Aug. to 10 Sept. Here also the scale could be inversed without having to change the signs (heave as negative LOS values but upwards).

- Figures 6-7-8-10: Use cm instead of m, as you wrote and mapped everything using cm.

- Figure 6: Not clear what is the point having a) and b) as they are basically showing the same, as also described at l.200. Locations a-c) are actually pts 1-3, I guess? And they are shown in 7b, not 7d.

- Figure 7: to me, Figure 7a (with Figure 8) is the best part of your article. Especially

the grey part with the indeterminate is great. It is a bit unfortunate that it is visually the part that we see the least (due to contrast).

- Figure 9: a) has also poor contrast: maybe because of the color choice it looks like a semi-transparent mask has been applied. b) topography would maybe be more useful as background. NIR-R-G does not bring much info here.

——

Potential for additional maps:

- It is missing somewhere a map showing the total subsidence or the early-season subsidence only compared to late-season, as you are discussing it in the text. Maybe as supplement?

- As you show examples of time series in Figure 10 that are estimated as ice-poor/ice-rich pixels based on InSAR subsidence in intermediate mapped areas, why not do the exercise at full scale and finish the article showing a whole map ice-poor/ice-rich. Of course, it would be an indicator/proxy/estimate, with some uncertainties/limitations (well explained in section 5.2), especially in areas where the distribution overlaps (where it should probably remain 'undefined'). But that would beautifully close the loop, I think.

——

Discussion:

This is a bit a mismatch between 5.2.2 and 5.3. With almost too negative statements in 5.2.2. (for ex l.280: 'late-season subsidence in a warm summer is not a perfect indicator of whether the top of the permafrost is ice rich or not') and almost too positive ones in 5.3 (for ex l. 310-312: 'late-season subsidence can be enhance the automated mapping of vulnerable permafrost ground ice' and 'the mapping can be automated, as no manual interpretation and no calibration using in-situ cores are required'). I understand the reason of both statements, but just think the text needs to be slightly more

balanced. For example, in 5.2.2, it is obviously good to discuss the limitations but somehow the pretty negative first sentence is rather confusing at this stage. Consider rephrasing and starting by saying where/when it is likely to work (extremely warm summers when the thaw front penetrates substantially into the top permafrost or where the ice content at the very top is abundant) and then discuss the problems. In 5.3. as you say that incorporating geological constraints can counteract weaknesses (l.319), it is probably good to change a bit the sentences at l.312. It may be technically possible to automate but that does not fully replace the needs for manual interpretation and in-situ measurements. In general, I would say the main point is: ground ice maps based on remotely sensed subsidence are complementary to other techniques and can contribute to upscale the identification of potentially hazardous areas.

–––––––––––––––––––––

Complementary comments:

–––––––––––––––––––––

- L.35-36: could be rephrase to: . . . that the mapping identifies degradational features when it is already too late.

- In section 2: missing an information about active layer thickness. Documented in this region? What is the typical thickness and which variability? Would be a useful information when discussing the limitations in section 5.2.2.: one way to identify false negatives may be to estimate which subsidence is expected from the 'normal' thawing of the active and transient layers.

- L.74: missing reference.

- L.82: heavy sentence. Not clear what is 'with respect to the previous decade'

- In 3.1.1. Could mention somewhere that this is based on images acquired with an ascending geometry. LOS arrow could be shown on maps.

- L.94: ref to Copernicus Sentinel should come in 3.1.1

- L.98: which DEM resolution?

- L.99: which multi-looking factors, and so which ground resolution?

- L.108: maybe mention here already that t1 actually is 10 August.

- L.123: this is the first (maybe only?) mention of the time period used to define with is late-season. Could be useful to remind it e.g. in figure legends, in conclusion. Maybe in the abstract as well.

- In 3.1.3: ref to figure where we see the points.

- In 3.1.4: What about the inter-annual variations in timing? Here the start-end of the late-season are fixed to 10 August / 10 September. May it become a problem when thinking about automation and processing of many years? It could perhaps be mentioned here or in the discussion.

- L.134: 'the sensitivity and specificity of the late-season subsidence indicator. . .' or 'the sensitivity and specificity of the late-season subsidence for ground ice mapping. . .'

- L.139: which percent is not mapped due to lack of unambiguous indicators? Could be mentioned as for the discarded areas, cause what we actually care is to know the percent that is documented (vs what is not in total).

- L.167: $ -> &

- L.169: I believe that Tatchim Isua has not been introduced yet so more info about where it is and a reference to a map would be welcome.

- L.187: weird to say 'peaks' when referring to a map. What about: 'The distribution of the observed late-season subsidence in our study area shows two distinct ranges of values'?

- L.188: later you mention 5-8 cm (e.g. l.196). Good to be consistent.

- L.216: ref to Supplement table?

- Figure 10: legend: '. . . from 2019 for point from Figure 7b).'

- L.225-231: this is especially where I thought: where is the final full-scale map showing the estimated distribution of ice rich / ice poor areas based on subsidence observations?

- L.243: 5-8 cm? A number 10 probably misplaced after 'late-season subsidence'

- L.254-259: this part if already focusing on limitations. Could be moved to 5.2.

- L.328: here comes another version of the range of values: 4-8 cm...

- L.338: could mention here again the likely difficulties in other kinds of environments (discontinuous permafrost, more vegetated).

- L.345: this point starts with 'Its'. Do not know what it is referring to. Points 4 and 5 could be merged.

---

## Referee Comment (RC2) · Anonymous Referee #2 · 3 Dec 2020

This manuscript shows an original interpretation of InSAR data by isolating the late-season subsidence of an extremely warm summer. Authors then link the high late-season subsidence, significantly different from the low late-season subsidence of ice-poor permafrost, to the degradation of the ice-rich top of permafrost. Although, there are limitations to this approach (well addressed by the authors), there is also a great value to spread the concept so the same idea can be tested in different permafrost environments, and perhaps in more complex settings. I support having this paper published in The Cryosphere. In general, the paper reads well, however, some suggestions are made for clarifications and additional information that can help improve the understanding and support the conclusion.

[Figure]

Below, I summarize my main comments, followed by specific comments and comments on Figures.

Main comments:

1. Transient layer versus intermediate layer

From the start (abstract), it would be important to bring the distinction between the transient and the intermediate layers (ice-rich top of permafrost) and to explicitly say to what the late-season subsidence is related to (ice-rich top of permafrost). There is some confusion in the introduction about this (lines 38-45 and line 55) and while you mainly associate the late-season subsidence to melting excess ice at the top of permafrost, you can't exclude the thawing of the transient layer as a contributing factor (as mentioned in your discussion). If possible, you should try to provide a description of the ice content within the transient layer as observed in the cores. In doing so, you can perhaps estimate the subsidence associated to the thawing of the transient layer and compare this estimate with the magnitude of late-season subsidence derived from Sentinel-1. This will give you arguments to support your conclusion.

2. Selection of the beginning of the late-season subsidence

The beginning of the late-season subsidence is mentioned quickly in the text (and too late, in line 123) and it is embedded in the caption of Figure 1. The definition of this late-season (especially the beginning) is central to your results and you should discuss your choice of selecting the second week of August for your study area. Your late-season is quite large (1 month) and likely encompass thawing of the active layer, transient layer and ice-rich top of permafrost. Furthermore, for automated ground ice mapping, selection of the late-season period will likely vary across the North.

3. Resolution of 60 m and detection of subsidence associated to the degradation of ice wedge polygons

At several places in the manuscript, you make the association of ice-rich permafrost to
the presence of ice wedge polygons; more precisely, you classify ice-rich permafrost based on visible ice wedges, but what about the ground ice content (excess ice) in the center of the polygons? My concern is about the resolution of 60 m used to derive the subsidence. Your manual classification of the ice-rich class seems to be based on visible ice wedges, but the InSAR result returns a 60 m pixel value where the subsidence is a mixture of various surfaces.

4. Result over the Kivalina study area

I understand that you want to present a simple case (focus area) to illustrate your approach. However, although not available for review, you seem to have, for your study area (Kivalina), the ground ice map in term of ice-rich, ice-poor, and indeterminate classes. Why then you did not provide the assessment for your entire study area?

Specific comments:

Line 3 : Could you define the late-season period?

Line 5 : Make the distinction between the transient layer and the intermediate layer (ice-rich top of permafrost).

Line 6: Please consider rewording "For locations independently determined to be ice rich" which is a little hard to understand at this stage.

Line 7: Is it also the 5th–95th percentile for the range of subsidence of ice-poor areas? Please add if so.

Lines 25-27: "Current approaches for mapping ground ice have significant shortcomings. One approach, palaeogeographic modelling of ground ice aggradation and degradation, is currently limited to coarse-scale assessments (Jorgenson et al., 2008; O'Neill et al., 2019)". True, but in fact, it is not the approach that is limited, but the input data that limits the result of such approach (in particular, the scale of the surficial geology used. In O'Neill et al. 2019, the surficial geology is at 1:5000000. As stated by O'Neill et al. 2019, the model output could be improved by including updated surficial geology

mapping). Please consider rewording.

Lines 38-45: It is somehow understood here that you relate the late-season subsidence to thawing of ice-rich top of permafrost (intermediate layer) and not to the thawing of the transient layer. However, you can be clearer by adding the transient layer and intermediate layer in your schematic of Figure 1 and by explicitly say to what the late-season subsidence is related to.

Line 55: In contrast to previous statement (in lines 38-45), you seem here to relate late-season subsidence to both "excess ice at the base of the active layer and top of the permafrost". Please clarify.

Line 71: Please define the rubble-covered surfaces in term of material or terrain unit type.

Line 71: Please give the mean annual ground temperature of this warm permafrost and the range of active layer thicknesses with the approximate date of the maximum thaw front (see comment of line 108).

Lines 78-79: "Ice-rich layers of segregated ice are also ubiquitous in fine-grained sediments (Shannon & Wilson, Inc., 2006)" Do you mean in marine sediments? Please clarify.

Line 82 and Fig 2b: Please add the climatic normal (TDD) in Fig. 2b, so that the decade and summer 2019 will be put into a longer perspective.

Lines 80-85: Could you also define the climate of the three years (or summers) in term of precipitations? That may help understand the soil moisture content at the end of the summers and support some discussion points (e.g. soil moisture used to aggrade ice at the base of the active layer, line 298). In addition, Douglas et al. (2020) recently shown the relationships between anomalously wet summers and thaw depth in discontinuous permafrost (Alaska). Therefore, not only extremely warm summers can lead to degradation of vulnerable excess ice in the upper permafrost, but extremely wet

summers as well.

Lines 89-92: Please add the resolution of Sentinel-1 interferometric wide mode. The resolution of 60 m is obtained by multi-looking (lines 98-99). This might be obvious to remote sensing experts, but not necessarily to permafrost experts. Please explain your choice for the resolution of 60 m.

Lines 89-92: Could you add the looking direction of the satellite?

Lines 98-99: Perhaps, briefly explain the purpose of multi-looking or link the multi-looking to the 60-m resolution.

Line 106: Can you show that there is no aspect-dependent trends that are associated with downslope movements in the supplementary file?

Line 108: Could you define and discuss your choice for the beginning of the late-season subsidence in the text. In Fig 1, you mention second week of August. When is the maximum thaw front generally reached? Please add this information in the study area section (see comment of line 71).

Lines 113-115: Please refer to Fig.3b

Line 123: If not done previously, please explain why you choose August 10 as the starting point of your late-season. Your late-season is quite large (1 month) and likely encompass thawing of the active layer, transient layer and ice-rich top of permafrost. Please explain.

Line 132: Please add a reference to your ground ice map. However, and unfortunately, the independently derived ground ice map (Zwieback, 2020a) is not available for review (see comment of lines 515-516).

Line 155: Alluvial deposit seems to be classify as ice-rich based on visible ice wedge polygons, what about the ground ice content in the center of the polygons, can you add this information?

Line 168: Can you specify the surficial geology of all sites in Shannon & Wilson, Inc., 2006 since the readers don't have access to this reference? Perhaps add the information in Table S1.

Lines 168-170: Again, does ice-rich permafrost associated to ice wedge polygons only?

Line 175: Even if you acknowledge that the cores were taken in 2005 compared with Sentinel-1 over summers 2017-2019, you should try to provide a description about the ice content of the transient layer as observed in 2005.

Line 182: Please correct Table S1, it should be Cores from 2005, not 2015.

Lines 179-181: Could you specify if ground ice contents represent the first meter below top of permafrost? Perhaps add the depth interval corresponding to the ground ice description in Table S1.

Line 187: Not sure if the use of the word "peak" to describe spatial variability is appropriate since it can be confused with a temporal peak. Please consider rewording.

Line 189: For negative late-season subsidence, do you mean in 2017 or 2019 or both?

Line 189: Do you mean uplift displacement and/or displacement toward the satellite? Please clarify. Did you add this sentence to support lines 105-106 about downslope movement? Please clarify.

Line 200: Again, it looks like the ice-rich deposits are only associated to the presence of ice wedge, and therefore, associating the subsidence to ice wedge degradation, however, at a resolution of 60 m, subsidence will rather reflect the one in the polygon center or at least be an average of polygon center and trough.

Line 207: In your manuscript, you chose to shown only the results over the limited focus area. However, your manual classification into ice-rich and ice-poor terrain (and indeterminate category) was done for the study area of Kivalina (Zwieback, 2020a –

unfortunately unavailable). It would have been interesting (perhaps at the end of your result section) to provide the distribution of the late-season subsidence over the Kivalina study area, such that someone can appreciate the performance of the approach compared to a small, almost ideal case study.

Line 207: Even if the difference in 2017 between ice-rich and ice-poor areas is smaller than 2019, is it statistically different?

Line 209: Is it also the 5th–95th percentile for the range of subsidence of ice-poor areas? Please add if so.

Lines 211-213: Should you also discuss 2017 and 2018? You can perhaps quickly say that the suitability of cooler years will be address in section 4.2.2

Line 234: What the distribution overlap in 2017? Even with large overlap, is it statistically different?

Lines 220-224: See comments for Figure 10.

Line 248: It is true that the transient layer contains less ice than the underlying intermediate layer (Shur et al., 2005) commonly known as the ice-rich top of permafrost. However, the transient layer could contain excess ice that will lead to late-season subsidence (you refer to this in lines 297-298). From the coring done in 2005, is there any indication of the thickness of the transient layer and its ice content? If yes, you can perhaps estimate the subsidence of the transient layer and compare with the magnitude of late-season subsidence derived from Sentinel-1.

Line 274: Please specify the magnitude of this spurious heave signal?

Lines 282-283: Do you infer than the warm summer of 2018 was enough to thaw the transient layer? Please clarify.

Lines 297-298: See comment of line 248 about the ice content of the transient layer from the 2005 cores.

Line 310: This automated ground ice mapping presuppose the selection of a starting date for the late-season subsidence; this starting date (and the end) will vary across the North. As mentioned previously, you should consider adding a discussion on the choice, for your study area, of the starting date for the late-season subsidence.

Line 312: However, frozen cores will be needed for the interpretation of ground ice content (e.g., in the transient layer, not only to estimate excess ice at depth – line 321) as well as other ground truthing to reduce observational uncertainties. Please consider nuancing your sentence and moving lines 319-322 after the first paragraph.

Line 316: How, with the resolution of Sentinel-1, the results can show subsidence of ice wedges?

Lines 515-516: These two products are unavailable to review:

Zwieback, S.: Kivalina ground ice map (Version 1.0), https://doi.org/10.5281/zenodo.4072407, 2020a. Zwieback, S.: Kivalina subsidence observations (Version 1.0), https://doi.org/10.5281/zenodo.4072257, 2020b.

Is it possible to consider adding them in the supplementary file?

Figures:

General: I suggest changing the color of your late-season subsidence scale; the blueish color is not the best to see the contrast, everything looks the same. You can maybe try the brown to blue, you used at Figure 5a-b, but only for subsidence (positive values). This scale will also be consistent with the colors chosen at Figure 7c, and choose another scale of colors for any heave displacement (negative values in Figure 5 a-b).

Figure 1: Please consider adding the transient layer and intermediate layer in your schematic of Figure 1.

Figure 2: Please add a horizontal line corresponding to the climatic normal; 30-year
average in term of TDD.

Figure 3: Not sure if Fig. 3a and 3b go together. The figure title does not really represent Fig. 3b? Could they be separate figures?

Figure 4: Please locate these two sub-regions within Figure 4 or Figure 2 and add an arrow for the North. Also, please define Catena.

Figure 5: Please add the reference of Figure 2 for the study area shown in Figure 5 "Regional variability of remotely sensed late-season subsidence dl within the study area (see Figure 2 for location)" Figure 6: Move this Figure after its first mention in the text, after line 199.

Figure 7: Please add the reference of Figure 5 for the focus area shown in Figure 7 "...determined by manual mapping in the focus area (see Figure 5 for location)"

Figure 7: Please consider rewording "a) Versus independent ground ice map and cores" (it took me some time to understand the meaning of that sentence). Suggestion: For example, "a) Distributions of late-season subsidence in 2019 according to the three ground ice classes manually/independently mapped"

Figure 7: I like Fig. 7a, but I don't understand why the distribution has to be in both direction, can it just be a "positive" distribution? I was trying to understand the "lense" effect.

Figure 7: "The diamonds indicate points mentioned in the text" this is rather vague, could you say "for points mentioned in Figure 10"?

Figure 8: I like this figure!

Figure 9: Please refer to Figure 5 for site location (after the mention of Tatchim Isua site).

Figure 10: Why points 4 to 19 while in Fig 7b-d it is points 1 to 17 with points 11, 18, and 19 missing? Please explain.

Figure 10: point 4, independently determined as indeterminate not ice rich, please explain or correct.

Figure 10: point 17, looks more ice poor than indeterminate according to manual mapping, please explain or correct.

Reference:

Douglas, T.A., Turetsky, M.R. and Koven, C.D. 2020. Increased rainfall stimulates permafrost thaw across a variety of Interior Alaskan boreal ecosystems. Climate and Atmospheric Science (2020) 3:28, https://doi.org/10.1038/s41612-020-0130-4

---

## Referee Comment (RC3) · Anonymous Referee #3 · 23 Dec 2020

Review of "Vulnerable top-of-permafrost ground ice indicated by remotely sensed late-season subsidence" by Simon Zwieback and Franz Meyer

This work proposed the use of late-thaw-season subsidence occurring in an extremely warm year to indicate the presence of ground ice at the top of permafrost. The study builds on a simple idea that if ice-rich ground thaws at the end of a warm summer, the resulted surface subsidence should be larger than in other normal or cooler summers. The authors presented InSAR observations in their study area that soundly support this idea and further backed up using independent ground ice mapping as obtained based on borehole data and manual interpretation from optical imagery. Overall, this

is an innovative way to utilize InSAR data for studying permafrost. Despite that the current work only provides a new indicator for the presence of excess ground ice at near surface, it may lead to more quantitative estimates of ice content or even temporal changes of ice content using observations alike.

I would like to raise a few points, hoping to improve the clarity and rigor of the paper.

1. Excess ground ice

For TC readership, it would be better to first define what excess ground ice is, e.g., referring to the Glossary of permafrost and related ground-ice terms as 'the volume of ice in the ground which exceeds the total pore volume that the ground would have under natural unfrozen conditions'. A clear definition would help to interpret the schematics as shown in Figure 1 and equation 1. Since the excess is relative to pore volume, it is more rigorous to include the contribution of pore ice. The authors' strategy is clever as the use of late-thaw-season subsidence in a warm summer implicitly removes the contribution from melting of pore ice (which happen in all summers, manifesting in the 'early-mid-season' results shown in Figs 6 and 10). The validity of equation 1 also builds on the fact that subsidence due to the melting of pore ice is negligible in late summer.

Figure 1 is confusing at first sight. The ice content profiles imply excess ice in the active layer, even in the ice poor case (or the authors mean pore ice instead of excess ice in the active layer?). And a minor note is that the label 'heave - subsidence' unnecessarily implies heave. The figure caption is already clear.

2. Top of permafrost

The classic two-layer model, active layer on top of permafrost, is adopted and well suits the nature of this work. In the discussion section, the authors did introduce the more complete four-layer structure: active layer, transient layer, intermediate layer, and permafrost. As the time scale of concern is an extremely warm summer within a few

years (three, in this case), it is very likely that the excess subsidence is due to melting of ice-rich intermediate layer or transient layer (later of which is possibly ice rich, yet the authors claimed to be ice poor, L248), instead of the permafrost below. Then, is it justifiable to claim that excess ground ice is present at the top of permafrost (part of permafrost)? I would be more comfortable to use 'near permafrost table' or alike to allow some leeway. Quantitively, excess subsidence of 5 cm may correspond to an ice-rich layer of 5-10 cm, roughly the same order of magnitude as the thickness of transition zone (transient layer + intermediate layer).

3. Extremely warm summer

I have no doubt that 2019 was an extremely warm summer in Kivalina (it was the warmest according to Fig. 2b). Without a statistical or meteorological perspective, I would also regard 2018 as very warm (2nd warmest in the Fig 2b time series); yet the late-summer subsidence in 2018 was normal. Then I was wondering how 'exceptional' the warming must be to cause the excess subsidence. Is it ever possible that the excess subsidence in 2019 resulted from the decadal warming in the region, esp. considering that it typically takes decades or longer to thaw permafrost in continuous permafrost zones?

4. Timings of late thaw season

The choices of the beginning and end of late season (namely, t1 and t2) make sense. I was just wondering if these are backed up by temperature data and do you need to adjust them when applying this method in different areas (I suppose you need). The authored mentioned 'diurnal frost heave', were there any signs of heave in September? Then how about late-summer heave (Mackay 1983 10.1139/e83-012)?

5. Overall, I think larger-than-usual late-season subsidence is an indirect, instead of direct, indicator of "vulnerable excess ground ice" near the permafrost table.

Minor comments

3.1.4 The authors may provide more details of the spline fitting and help us to understand Fig 3b.

L167: $ should be &

L209: Delete the extra for

L228: indicate should be indicates

L251: Its should be It

L312: an should be and

L316: includes should be include

---

## Author Comment (AC1) · 19 Jan 2021

Zwieback and Meyer studied the suitability of exploiting InSAR-based late-season subsidence measurements in northwestern Alaska as an indicator of excess ground ice at the top of permafrost. They present a great piece of work that shows the value of radar remote sensing to upscale ground ice mapping and identify potentially unstable terrain at large scale. This topic is relevant both for Arctic research and operational end-user exploitation. The paper is well suitable for a publication in The Cryosphere and likely to become a reference in remote sensing of permafrost. I have no major concern regarding the methods and main results, but I think the manuscript could benefit from some additional information, clarifications, and adjustments, especially in figures. These elements, so called 'main comments', are described thereafter. I also listed more technical suggestions, so called 'complementary comments', in the second part of the review.

We are grateful to the referee for their helpful comments and suggestions. We address them below; we have grouped several closely related points.

Abstract

- Some parts of the current version are not easy to understand without having read the whole paper (although this should be the point of an abstract). For ex l.6 'For locations independently determined to be ice rich': Maybe sth like 'Compared to an independently-generated manual mapping of ice-rich and ice-poor areas, . . .';
l.7-8 'The distributions overlapped by 2%...': the first time I read it, I understood 'spatial distribution' which obviously did not make sense. Maybe sth like: 'the distributions of the late-season subsidence values in referenced ice-rich/ice-poor areas overlap by only 2%, which. . .'.
- Seems that it was a conscious decision all along the article not mentioning that the subsidence results are InSAR-based. I understand that the detection technique is not the main point here but still think it would be good to mention it a couple of times, at relevant locations. For instance, in the abstract (l.3) and conclusions (l.326).- Somehow l.58-66 do a better job to clearly summarize the work. Could maybe be used to rework the abstract?

We agree with these suggestions. We have expanded the description of the independently derived ground ice information and clarified that it is the statistical distributions that overlap. We have added the word InSAR to the abstract and the main body of the paper. Finally, the last sentence now establishes a tighter link between this study and the ultimate goal of anticipating terrain instability.

Figures: I believe some figures could become easier to read with minor changes. Here are some suggestions:

- Figure 1 right: A bit confusing that subsidence is written on the left (y-axis) but subsidence-heave is indicated on the right (in addition with counter-intuitive directions). Indicate that the lack horizontal line in the center corresponds to 0 and maybe consider inversing the direction (heave upwards, subsidence downwards). Except if there is really a good reason not doing so, it would be much easier to read. Positive sign mentioned in the legend is anyway not shown in the figure and I guess (+/-) here is used to correspond to changes of sensor-to-ground distance, but it should not impact the visualization (if you imagine Sentinel-1 at the top of the page and the ground at the location of your horizontal straight line, an increase of distance goes downwards). Consider also having all text in black instead of light grey.

We will tweak the layout to make the figure more readable. The reason for showing subsidence as positive throughout the manuscript and thus also in this figure is that virtually all the movements we observed and those we are interested in are downward. Reversing the sign convention would make for rather awkward verbal descriptions.

> - Figure 2 left: What do the different colors of the spline lines actually indicate? Missing a legend.

We will add that their purpose is to make the lines easily distinguishable.

> - Figure 4: Maybe add an information about location. In legend: what does Catena mean here? Is it a place?

We will fix this and replace the term catena by hillslope sequence.

> - Figure 5: As you refer to 5-8 cm/yr of typical values, I wonder if your subsidence maps would not benefit from a better contrast by reducing min/max values of your color scale (-10 to 10?). Or considering other colors or an asymmetric scale (I guess you do not have heave values up to -15). Add in legend somewhere that blue = no data in a) / b), i.e. under a certain coherence threshold I guess (which btw is not mentioned).Remind somewhere that late-season is 10 Aug. to 10 Sept. Here also the scale could be inversed without having to change the signs (heave as negative LOS values butupwards).

We will experiment with the contrast to improve the figure and include the additional information (missing data, definition of the late season). We are aware of the unintuitive nature of our subsidence sign convention, but as stated above and in the paper, we believe that it simplifies the discussion. We will try flipping the scale but it may complicate things further.

> - Figures 6-7-8-10: Use cm instead of m, as you wrote and mapped everything using cm.

Agreed.

> - Figure 6: Not clear what is the point having a) and b) as they are basically showing the same, as also described at l.200. Locations a-c) are actually pts 1-3, I guess? And they are shown in 7b, not 7d.

Fixed.

> - Figure 7: to me, Figure 7a (with Figure 8) is the best part of your article. Especially the grey part with the indeterminate is great. It is a bit unfortunate that it is visually the part that we see the least (due to contrast).

Thank you. We have modified the brightness to make it stand out a more.

> - Figure 9: a) has also poor contrast: maybe because of the color choice it looks like a semi-transparent mask has been applied. b) topography would maybe be more useful as background. NIR-R-G does not bring much info here.

We will improve the figure by tweaking the contrast and the scale. We also want to visually identify the ice-rich and ice-poor cores more clearly. Conversely, we think the satellite image provides more information than elevation in this case, because of the close correspondence between the visual appearance (largely due to variations in vegetation cover) and the surficial geology/permafrost conditions.

Potential for additional maps:

- It is missing somewhere a map showing the total subsidence or the early-season subsidence only compared to late-season, as you are discussing it in the text. Maybe as supplement?

We agree and will add a figure showing the early-season subsidence in the three years to the supplement.

- As you show examples of time series in Figure 10 that are estimated as ice-poor/icerich pixels based on InSAR subsidence in intermediate mapped areas, why not do the exercise at full scale and finish the article showing a whole map ice-poor/ice-rich. Of course, it would be an indicator/proxy/estimate, with some uncertainties/limitations (well explained in section 5.2), especially in areas where the distribution overlaps (where it should probably remain 'undefined'). But that would beautifully close the loop, I think.

This is a good but somewhat delicate point. We agree that adding such a result (and briefly explaining the rationale for choosing a particular threshold) has the potential to strengthen the manuscript. The main caveat is that there is a risk of misrepresenting the readiness of these kinds of observations for ground ice mapping. Our main goal remains to assess the suitability of late-season subsidence for this task, rather than validating a particular classification algorithm based on these data.

We believe an expedient way to achieve these goals is to add a panel to Fig. 7 and keep the discussion of this result focused.

Discussion:

This is a bit a mismatch between 5.2.2 and 5.3. With almost too negative statements in 5.2.2. (for ex l.280: 'late-season subsidence in a warm summer is not a perfect indicator of whether the top of the permafrost is ice rich or not') and almost too positive ones in 5.3 (for ex l. 310-312: 'late-season subsidence can be enhance the automated mapping of vulnerable permafrost ground ice' and 'the mapping can be automated, as no manual interpretation and no calibration using in-situ cores are required'). I understand the reason of both statements, but just think the text needs to be slightly more balanced. For example, in 5.2.2, it is obviously good to discuss the limitations but somehow the pretty negative first sentence is rather confusing at this stage. Consider rephrasing and starting by saying where/when it is likely to work (extremely warm summers when the thaw front penetrates substantially into the top permafrost or where the ice content at the very top is abundant) and then discuss the problems. In 5.3. as you say that incorporating geological constraints can counteract weaknesses (l.319), it is probably good to change a bit the sentences at l.312. It may be technically possible to automate but that does not fully replace the needs for manual interpretation and in-situ measurements. In general, I would say the main point is: ground ice maps based on remotely sensed subsidence are complementary to other techniques and can contribute to upscale the identification of potentially hazardous areas.

We are grateful for the suggestions and plan to improve these two sections as follows:

- More gentle and balanced introduction to section 5.2.2

  The high suitability of late-season subsidence for mapping ice-rich permafrost that we identified in the study area was arguably promoted by the exceptionally warm summer of 2019 and the high ice content at the top of permafrost. In less propitious circumstances, its sensitivity and specificity may be reduced.

- We have restructured section 5.3 in an attempt to make the writing clearer and more balanced. In particular, in the first paragraph (formerly starting at l312):

    Late-season subsidence can enhance the automated mapping of vulnerable permafrost ground ice. Remotely sensed late-season subsidence can be mapped on pan-Arctic scales, thanks to the global availability of Sentinel-1 data. A further practical advantage is that it lends itself to automation, as no manual interpretation and no calibration using in-situ cores are required. To automate the specification of the late-season period, globally available reanalysis data could be considered. Despite the potential for automation, we believe that the greatest potential lies in its synergistic use with expert knowledge, field observations and existing mapping approaches.

    Complementary comments

We are grateful to the referee for these helpful comments. We have fixed the errors and inconsistencies. We only respond to the more substantial comments.

   - L.35-36: could be rephrase to: . . . that the mapping identifies degradational features when it is already too late.

Rephrased to "that identifying ice-rich permafrost using degradational features works best when it is already too late".

   - In section 2: missing an information about active layer thickness. Documented in this region? What is the typical thickness and which variability? Would be a useful information when discussing the limitations in section 5.2.2.: one way to identify false negatives may be to estimate which subsidence is expected from the 'normal' thawing of the active and transient layers.

We have added:

    While no contemporary active layer thickness measurements are available, Shannon & Wilson report values ranging from 0.5 to 1.0 m.

   - L.74: missing reference.
   - L.82: heavy sentence. Not clear what is 'with respect to the previous decade'
   - In 3.1.1. Could mention somewhere that this is based on images acquired with an ascending geometry. LOS arrow could be shown on maps.

Fixed.
We have clarified that the flight direction was descending (Sentinel-1 is a right-looking satellite).

   - L.94: ref to Copernicus Sentinel should come in 3.1.1
   - L.98: which DEM resolution?
   - L.99: which multi-looking factors, and so which ground resolution?
   - L.108: maybe mention here already that t1 actually is 10 August.

Agreed

   - L.123: this is the first (maybe only?) mention of the time period used to define with is late-season. Could be useful to remind it e.g. in figure legends, in conclusion. Maybe in the abstract as well.

Added to the first figure caption; the extent of the late season period is further shown in all time series plots.

   - In 3.1.3: ref to figure where we see the points.

We now link to the section in the figure caption (rather than the other way round).

> - In 3.1.4: What about the inter-annual variations in timing? Here the start-end of the late-season are fixed to 10 August / 10 September. May it become a problem when thinking about automation and processing of many years? It could perhaps be mentioned here or in the discussion.

We plan to add a separate sensitivity analysis in which we vary the beginning of the late season.

> - L.134: 'the sensitivity and specificity of the late-season subsidence indicator. . .' or 'the sensitivity and specificity of the late-season subsidence for ground ice mapping. . .'
> - L.139: which percent is not mapped due to lack of unambiguous indicators? Could be mentioned as for the discarded areas, cause what we actually care is to know the percent that is documented (vs what is not in total).

Agreed. We added that 19% of the area were assigned to the indeterminate category during the manual mapping.

> - L.167: $ -> &
>
> - L.169: I believe that Tatchim Isua has not been introduced yet so more info about where it is and a reference to a map would be welcome.
>
> - L.187: weird to say 'peaks' when referring to a map. What about: 'The distribution of the observed late-season subsidence in our study area shows two distinct ranges of values'?
>
> - L.188: later you mention 5-8 cm (e.g. l.196). Good to be consistent.
>
> - L.216: ref to Supplement table?
>
> - Figure 10: legend: '. . . from 2019 for point from Figure 7b).'
>
> - L.225-231: this is especially where I thought: where is the final full-scale map showing the estimated distribution of ice rich / ice poor areas based on subsidence observations?
>
> - L.243: 5-8 cm? A number 10 probably misplaced after 'late-season subsidence'
>
> - L.254-259: this part if already focusing on limitations. Could be moved to 5.2.
>
> - L.328: here comes another version of the range of values: 4-8 cm...

We will amend all of these points.

> - L.338: could mention here again the likely difficulties in other kinds of environments
>
> (discontinuous permafrost, more vegetated).

Highlighted boreal forest

> - L.345: this point starts with 'Its'. Do not know what it is referring to. Points 4 and 5
>
> could be merged.

Clarified the sentence. We will consider merging the two points.

---

## Author Comment (AC2) · 19 Jan 2021

This manuscript shows an original interpretation of InSAR data by isolating the lateseason subsidence of an extremely warm summer. Authors then link the high lateseason subsidence, significantly different from the low late-season subsidence of icepoor permafrost, to the degradation of the ice-rich top of permafrost. Although, there are limitations to this approach (well addressed by the authors), there is also a great value to spread the concept so the same idea can be tested in different permafrost environments, and perhaps in more complex settings. I support having this paper published in The Cryosphere. In general, the paper reads well, however, some suggestions are made for clarifications and additional information that can help improve the understanding and support the conclusion.

**We are grateful to the reviewer for their insight and constructive suggestions which we are confident will help us improve the clarity and balance of the manuscript.**

Below, I summarize my main comments, followed by specific comments and comments on Figures.

Main comments:

1. Transient layer versus intermediate layer

From the start (abstract), it would be important to bring the distinction between the transient and the intermediate layers (ice-rich top of permafrost) and to explicitly say to what the lateseason subsidence is related to (ice-rich top of permafrost). There is some confusion in the introduction about this (lines 38-45 and line 55) and while you mainly associate the late-season subsidence to melting excess ice at the top of permafrost, you can't exclude the thawing of the transient layer as a contributing factor (as mentioned in your discussion). If possible, you should try to provide a description of the ice content within the transient layer as observed in the cores. In doing so, you can perhaps estimate the subsidence associated to the thawing of the transient layer and compare this estimate with the magnitude of late-season subsidence derived from Sentinel-1. This will give you arguments to support your conclusion.

We agree and plan to extend the discussion of the transient and the intermediate layer. Having said that, we also note that a process-based interpretation of the satellite observations is not a major goal of ours. We emphasize this point in the discussion, where we talk about the need for contemporaneous and dense ground-ice samples in order to reliably interpret the observations and to assess the potential to quantify ice contents (i.e., to move beyond the two-class taxonomy of ice-rich and ice-poor terrain).

We have added a separate paragraph in the introduction.

The stratigraphy of permafrost-affected soils adds complexity to the link between upper-permafrost ice content and remotely sensed late-season subsidence. To describe the cryostratigraphy in ice-rich terrain, Shur et al. divide the long-term permafrost into three layers. The uppermost layer, the transient layer, generally has a low to moderate excess ice content, as a result of occasional deep thaw. Disappearance of the transient layer is frequently triggered by sustained warming or disturbance. The subjacent ice-rich intermediate layer is then exposed, increasing the susceptibility to enhanced subsidence. The risk of sustained thaw consolidation is amplified where the intermediate layer overlies massive ice such as ice wedges. Once the protection afforded by the transient and intermediate layer has been lost, further thaw will lead to ice wedge degradation. Ice wedge polygons also illustrate the large lateral variability in ground ice conditions, which need to be considered when interpreting late-season subsidence as an indicator of ice-rich upper permafrost.

We do not intend to change the figure because we feel these details would distract from the bigger picture.

We also plan changes to the discussion section. Most importantly, we intend to strengthen the discussion of how we suspect the transient layer contributed to the difference between 2018 and 2019:

Inter-annual variability in late-season subsidence of ice-rich areas poses challenges for ground ice mapping. Potential sources of inter-annual variability and trends include surface changes (e.g., soil moisture, disturbance, snow) and variable meteorological conditions such as precipitation. Memory effects could also be relevant. Taking the ice-rich area in Figure 8a) as an example, we speculate that thaw of materials with moderate excess ice contents (transient layer) at the end of the warm summer of 2018 (limited late-season subsidence of ~ 2 cm) could have promoted larger subsidence in 2019 by weakening the protection given to the subjacent materials richer in excess ice (intermediate layer, massive ice). Equally, the summer of 2018 may not have been warm (and wet) enough to allow for reliable identification of the vulnerable ground ice at this location. That the identification strategy presupposes an initial degradation of ground ice constitutes its biggest limitation.

2. Selection of the beginning of the late-season subsidence The beginning of the late-season subsidence is mentioned quickly in the text (and too late, in line 123) and it is embedded in the caption of Figure 1. The definition of this lateseason (especially the beginning) is central to your results and you should discuss your choice of selecting the second week of August for your study area. Your late-season is quite large (1 month) and likely encompass thawing of the active layer, transient layer and ice-rich top of permafrost. Furthermore, for automated ground ice mapping, selection of the late-season period will likely vary across the North.

We agree with these points. We plan to add a figure to the supplement where we show Fig. 8 for shorter periods. For this study area and the year 2019, the difference between the temporal trajectories of ice-rich and ice-poor terrain is, for the most part, sufficiently large so that we will not expect a substantial impact of the specification of the period on the separability. But this may not always be the case, and we will strengthen the analysis in the discussion and the methods.

3. Resolution of 60 m and detection of subsidence associated to the degradation of ice wedge polygons

At several places in the manuscript, you make the association of ice-rich permafrost to the presence of ice wedge polygons; more precisely, you classify ice-rich permafrost based on visible ice wedges, but what about the ground ice content (excess ice) in the center of the polygons? My concern is about the resolution of 60 m used to derive the subsidence. Your manual classification of the ice-rich class seems to be based on visible ice wedges, but the InSAR result returns a 60 m pixel value where the subsidence is a mixture of various surfaces.

We agree that this is a major complicating factor that we failed to address in sufficient depth in the manuscript. To remedy this failure, our planned major changes are:

• Include a separate bullet point on this issue in the description of the ground ice mapping:

A major caveat is that the presence of ice wedges provides no direct information on the ice content in the polygon interiors. However, previously taken cores (by Shannon and Wilson; to be discussed later}) from centres in various terrain units were ice rich, and the presence and ongoing expansion of thermokarst ponds and lakes also provides support for this assumption.

• Extend the discussion of the spatial variability

A big challenge, the sub-resolution spatial variability of ground ice, is exemplified by ice-wedge polygons. To what extent do the satellite observations reflect the wedges and the polygon interior, and how does it vary with factors such as ice content of the centres and the thickness distribution of the protective layer above the wedges? Quantitative answers will require densely sampled ground ice cores

• Further minor qualifications in the discussion, for instance

Site knowledge is indispensable for interpreting the stratigraphic complexity and sub-resolution variability of ground ice conditions

4. Result over the Kivalina study area

I understand that you want to present a simple case (focus area) to illustrate your approach. However, although not available for review, you seem to have, for your study area (Kivalina), the ground ice map in term of ice-rich, ice-poor, and indeterminate classes. Why then you did not provide the assessment for your entire study area?

We first apologize for uploading the data to the repository without granting access. This has been fixed.

The independent map was only derived for the focus area because of the laborious nature of the manual mapping (> 3 days for >  $60 \text{ km}^2$ ). In the methods section:

The mapped focus area, 8 km by 8 km in size, was chosen because of the wide range of ecotypes and the availability of field observations.

**Specific comments:**

**We have made numerous smaller changes. We have grouped several of these minor comments to streamline our response.**

Line 3 : Could you define the late-season period?

Line 5 : Make the distinction between the transient layer and the intermediate layer (ice-rich top of permafrost).

Line 6: Please consider rewording "For locations independently determined to be icerich" which is a little hard to understand at this stage.

Line 7: Is it also the 5th–95th percentile for the range of subsidence of ice-poor areas? Please add if so.

**We have clarified these points in the abstract. We prefer not to introduce the transient/intermediate layer model at this stage because we want to focus on the big picture.**

Lines 25-27: "Current approaches for mapping ground ice have significant shortcomings. One approach, palaeogeographic modelling of ground ice aggradation and degradation, is currently limited to coarse-scale assessments (Jorgenson et al., 2008; O'Neill et al., 2019)". True, but in fact, it is not the approach that is limited, but the input data that limits the result of such approach (in particular, the scale of the surficial geology used. In O'Neill et al. 2019, the surficial geology is at 1:5000000. As stated by O'Neill et al. 2019, the model output could be improved by including updated surficial geology mapping). Please consider rewording.

**Agreed.**

**Maps obtained from palaeogeographic modelling of ground ice aggradation and degradation, are currently limited to coarse scales**

Lines 38-45: It is somehow understood here that you relate the late-season subsidence to thawing of ice-rich top of permafrost (intermediate layer) and not to the thawing of the transient layer. However, you can be clearer by adding the transient layer and intermediate layer in your schematic of Figure 1 and by explicitly say to what the lateseason subsidence is related to.

Line 55: In contrast to previous statement (in lines 38-45), you seem here to relate late-season subsidence to both "excess ice at the base of the active layer and top of the permafrost". Please clarify.

**Please see our response to point 1.**

Line 71: Please define the rubble-covered surfaces in term of material or terrain unit type.

**Specified as well-drained uplands.**

Line 71: Please give the mean annual ground temperature of this warm permafrost and the range of active layer thicknesses with the approximate date of the maximum thaw front (see comment of line 108).

**Added range of active layer thicknesses and permafrost temperature.**

Lines 78-79: "Ice-rich layers of segregated ice are also ubiquitous in fine-grained sediments (Shannon & Wilson, Inc., 2006)" Do you mean in marine sediments? Please clarify.

Clarified that they occur throughout the study area, including uplands. We do not want to engage in excessive speculation about the quaternary history of the study area because the geological reports available to us focus on geotechnical properties. They do include interpretations as to the history of several locations, but the various reports are often in disagreement and, in our personal opinion, not always compelling.

Line 82 and Fig 2b: Please add the climatic normal (TDD) in Fig. 2b, so that the decade and summer 2019 will be put into a longer perspective.

Lines 80-85: Could you also define the climate of the three years (or summers) in term of precipitations? That may help understand the soil moisture content at the end of the summers and support some discussion points (e.g. soil moisture used to aggrade ice at the base of the active layer, line 298). In addition, Douglas et al. (2020) recently shown the relationships between anomalously wet summers and thaw depth in discontinuous permafrost (Alaska). Therefore, not only extremely warm summers can lead to degradation of vulnerable excess ice in the upper permafrost, but extremely wet summers as well.

We will add the precipitation observations that are available. It appears that 2019 was indeed a wet summer, but it is difficult to make comparisons because of the gaps the observations contain. We will also add the 30-year TDD average. We have further strengthened the discussion section by expanding the discussion of precipitation and soil moisture, and by adding references to the Douglas et al. paper and one by Shiklomanov et al. (continuous permafrost).

Lines 89-92: Please add the resolution of Sentinel-1 interferometric wide mode. The resolution of 60 m is obtained by multi-looking (lines 98-99). This might be obvious to remote sensing experts, but not necessarily to permafrost experts. Please explain your choice for the resolution of 60 m.

Lines 89-92: Could you add the looking direction of the satellite?

Lines 98-99: Perhaps, briefly explain the purpose of multi-looking or link the multilooking to the 60-m resolution.

**Done.**

Line 106: Can you show that there is no aspect-dependent trends that are associated with downslope movements in the supplementary file?

**We believe the existing figures are sufficient to show that there are no strong aspect-dependent trends.**

Line 108: Could you define and discuss your choice for the beginning of the lateseason subsidence in the text. In Fig 1, you mention second week of August. When is the maximum

thaw front generally reached? Please add this information in the study area section (see comment of line 71).

Lines 113-115: Please refer to Fig.3b

Line 123: If not done previously, please explain why you choose August 10 as the starting point of your late-season. Your late-season is quite large (1 month) and likely encompass thawing of the active layer, transient layer and ice-rich top of permafrost.

**Agreed. We have expanded this section, drawing attention to our new sensitivity analyses (see above) and spelling out the key considerations relevant to selecting the dates.**

Line 132: Please add a reference to your ground ice map. However, and unfortunately, the independently derived ground ice map (Zwieback, 2020a) is not available for review(see comment of lines 515-516).

**Done. We apologize that the map was previously inaccessible. This error has been fixed.**

Line 155: Alluvial deposit seems to be classify as ice-rich based on visible ice wedge polygons, what about the ground ice content in the center of the polygons, can you add this information?

**Agreed. See point 4.**

Line 168: Can you specify the surficial geology of all sites in Shannon & Wilson, Inc., 2006 since the readers don't have access to this reference? Perhaps add the information in Table S1.

We have extended the information provided in Table S1. We have added descriptions of the surficial geology, ice wedge polygons, topographic position and, where it could be classified with reasonable confidence, surficial geology.

Lines 168-170: Again, does ice-rich permafrost associated to ice wedge polygons only?

**Added clarification about where the cores were taken in relation to ice wedges.**

Line 175: Even if you acknowledge that the cores were taken in 2005 compared with Sentinel-1 over summers 2017-2019, you should try to provide a description about the ice content of the transient layer as observed in 2005.

Unfortunately, we are unable to do so because the report does not provide sufficiently granular information. For instance, for core 05-5, the soil below the thaw front is described in intervals of > 0.5 m (the top one reads: Organic SILT and ice-rich SILT; frozen.).

We plan to take our own cores next summer for more detailed studies.

Line 182: Please correct Table S1, it should be Cores from 2005, not 2015.

**Fixed**

Lines 179-181: Could you specify if ground ice contents represent the first meter below top of permafrost? Perhaps add the depth interval corresponding to the ground ice description in Table S1.

Unfortunately, the report does not contain measurements of excess or total gravimetric ground ice. Estimates of the visible ice content are not provided for all sections of all cores, even those that are ice rich (e.g., Organic SILT and ice-rich SILT; frozen; ICE with gray silt inclusions.). We have extended the description of the cores and their location in table S1 (see above). Line 187: Not sure if the use of the word "peak" to describe spatial variability is appropriate since it can be confused with a temporal peak. Please consider rewording.

**Fixed.**

Line 189: For negative late-season subsidence, do you mean in 2017 or 2019 or both?

**Clarified.**

Line 189: Do you mean uplift displacement and/or displacement toward the satellite? Please clarify. Did you add this sentence to support lines 105-106 about downslope movement? Please clarify.

**Simplified to "There were no notable instances of pronounced negative estimates in any of the years." to avoid invoking the slight ambiguity of the term subsidence.**

Line 200: Again, it looks like the ice-rich deposits are only associated to the presence of ice wedge, and therefore, associating the subsidence to ice wedge degradation, however, at a resolution of 60 m, subsidence will rather reflect the one in the polygon center or at least be an average of polygon center and trough.

We have removed the reference to the ice wedges, instead specifying that these locations were identified as ice rich in our independent map. We pick up the interpretation of the results in the discussion section.

Line 207: In your manuscript, you chose to shown only the results over the limited focus area. However, your manual classification into ice-rich and ice-poor terrain (and indeterminate category) was done for the study area of Kivalina (Zwieback, 2020a – unfortunately unavailable). It would have been interesting (perhaps at the end of your result section) to provide the distribution of the late-season subsidence over the Kivalina study area, such that someone can appreciate the performance of the approach compared to a small, almost ideal case study.

We first apologize for uploading the data to the repository without granting access. This has been fixed.

The independent map was only derived for the focus area because of the laborious nature of the manual mapping (> 3 days for >  $60 \text{ km}^2$ ). In the methods section:

The mapped focus area, 8 km by 8 km in size, was chosen because of the wide range of ecotypes and the availability of field observations.

Line 207: Even if the difference in 2017 between ice-rich and ice-poor areas is smaller than 2019, is it statistically different?

We are not entirely sure whether the referee is referring to the difference between years or between ice-rich and ice-poor areas. We avoid questions of statistical significance because the appropriate underlying population (to which the repetition intrinsic to significance testing refers) is not self-evident.

Line 209: Is it also the 5th–95th percentile for the range of subsidence of ice-poor areas? Please add if so.

Fixed.

Lines 211-213: Should you also discuss 2017 and 2018? You can perhaps quickly say that the suitability of cooler years will be address in section 4.2.2

**We will tweak this passage.**

Line 234: What the distribution overlap in 2017? Even with large overlap, is it statistically different?

**See line 207**

Line 248: It is true that the transient layer contains less ice than the underlying intermediate layer (Shur et al., 2005) commonly known as the ice-rich top of permafrost. However, the transient layer could contain excess ice that will lead to late-season subsidence (you refer to this in lines 297-298). From the coring done in 2005, is there any indication of the thickness of the transient layer and its ice content? If yes, you can perhaps estimate the subsidence of the transient layer and compare with the magnitude of late-season subsidence derived from Sentinel-1.

Unfortunately, the granularity of the reported results is insufficient for such purposes. Please see our response to the first point on how we improved the discussion of the transition zone.

Line 274: Please specify the magnitude of this spurious heave signal?

**Done.**

Lines 282-283: Do you infer than the warm summer of 2018 was enough to thaw the transient layer? Please clarify.

Lines 297-298: See comment of line 248 about the ice content of the transient layer from the 2005 cores.

**Please see our response to the first point on how we improved the discussion of the transition zone.**

Line 310: This automated ground ice mapping presuppose the selection of a starting date for the late-season subsidence; this starting date (and the end) will vary across the North. As mentioned previously, you should consider adding a discussion on the choice, for your study area, of the starting date for the late-season subsidence.

**We have added**

To automate the specification of the late-season period, globally available reanalysis data could be considered.

Line 312: However, frozen cores will be needed for the interpretation of ground ice content (e.g., in the transient layer, not only to estimate excess ice at depth – line 321) as well as other ground truthing to reduce observational uncertainties. Please consider nuancing your sentence and moving lines 319-322 after the first paragraph.

**We agree that the suggested structure works better. We further plan to add the following qualifying statement**

Incorporating geological constraints and expertise will be essential to counteract weaknesses of remotely sensed lateseason subsidence. Site knowledge is indispensable for interpreting the stratigraphic complexity and sub-resolution variability of ground ice conditions. Most importantly, observational constraints will be needed to estimate total excess ice contents. Line 316: How, with the resolution of Sentinel-1, the results can show subsidence of ice wedges?

**We have made several changes, albeit prior to this sentence. Please see point 4.**

Lines 515-516: These two products are unavailable to review: Zwieback, S.: Kivalina ground ice map (Version 1.0), https://doi.org/10.5281/zenodo.4072407, 2020a. Zwieback, S.: Kivalina subsidence observations (Version 1.0), https://doi.org/10.5281/zenodo.4072257, 2020b. Is it possible to consider adding them in the supplementary file?

**We apologize for not having made the data accessible. We have now made them publicly available.**

Figures

General: I suggest changing the color of your late-season subsidence scale; the blueish color is not the best to see the contrast, everything looks the same. You can maybe try the brown to blue, you used at Figure 5a-b, but only for subsidence (positive values). This scale will also be consistent with the colors chosen at Figure 7c, and choose another scale of colors for any heave displacement (negative values in Figure 5 a-b).

**We will adjust the color scales and the ranges to improve the contrast.**

Figure 1: Please consider adding the transient layer and intermediate layer in your schematic of Figure 1.

**We will experiment with it. Our concerns are that it may clutter the figure and exacerbate already existing issues to do with neglecting the spatial variability.**

Figure 2: Please add a horizontal line corresponding to the climatic normal; 30-year average in term of TDD.

**We will add it.**

Figure 3: Not sure if Fig. 3a and 3b go together. The figure title does not really represent Fig. 3b? Could they be separate figures?

**We agree that the juxtaposition is not ideal, but we contend that both pertain to how (accurately) lateseason subsidence can be extracted from the data.**

Figure 4: Please locate these two sub-regions within Figure 4 or Figure 2 and add an arrow for the North. Also, please define Catena.

**We will remove the word catena and mark the location of sub-regions.**

Figure 5: Please add the reference of Figure 2 for the study area shown in Figure 5 "Regional variability of remotely sensed late-season subsidence dl within the study area (see Figure 2 for location)"

**Agreed.**

Figure 6: Move this Figure after its first mention in the text, after line 199.

**Agreed.**

Figure 7: Please add the reference of Figure 5 for the focus area shown in Figure 7 ": : :determined by manual mapping in the focus area (see Figure 5 for location)" Figure 7: Please consider rewording "a) Versus independent ground ice map and cores" (it took me some time to understand the meaning of that sentence). Suggestion: For example, "a) Distributions of late-season subsidence in 2019 according to the three ground ice classes manually/independently mapped"

Figure 7: I like Fig. 7a, but I don't understand why the distribution has to be in both direction, can it just be a "positive" distribution? I was trying to understand the "lense" effect.

Figure 7: "The diamonds indicate points mentioned in the text" this is rather vague, could you say "for points mentioned in Figure 10"?

Figure 8: I like this figure!

We have amended the caption. In regard to the "lense" effect: we show the negative values in Fig. 7a, and also in Fig. 8, because they will need to be considered in any ground ice mapping effort. Here, the magnitudes of the negative values are generally within the observational uncertainty.

Figure 9: Please refer to Figure 5 for site location (after the mention of Tatchim Isua site).

Agreed.

Figure 10: Why points 4 to 19 while in Fig 7b-d it is points 1 to 17 with points 11, 18, and 19 missing? Please explain.

Time series for points 1-3 are shown in Fig. 6 (for all three years). Points 11, 18, 19 are outside the focus area and have now been added to Fig. 5.

Figure 10: point 4, independently determined as indeterminate not ice rich, please explain or correct.

Thank you for pointing out this error. We mixed up two different points in our database.

Figure 10: point 17, looks more ice poor than indeterminate according to manual mapping, please explain or correct

We agree that the fact that we assigned it to the indeterminate class is not readily apparent at this scale (not aided by the fact that the large marker obscures an area of substantial extent). The location is in the middle of a narrow strip of vegetated and soil-covered upland terrain that protrudes into the bedrock outcrop/rubble-covered ridge. We will consider choosing a different point to avoid this issue.

---

## Author Comment (AC3) · 19 Jan 2021

Review of "Vulnerable top-of-permafrost ground ice indicated by remotely sensed lateseason subsidence" by Simon Zwieback and Franz Meyer

This work proposed the use of late-thaw-season subsidence occurring in an extremely warm year to indicate the presence of ground ice at the top of permafrost. The study builds on a simple idea that if ice-rich ground thaws at the end of a warm summer, the resulted surface subsidence should be larger than in other normal or cooler summers. The authors presented InSAR observations in their study area that soundly support this idea and further backed up using independent ground ice mapping as obtained based on borehole data and manual interpretation from optical imagery. Overall, this is an innovative way to utilize InSAR data for studying permafrost. Despite that the current work only provides a new indicator for the presence of excess ground ice at near surface, it may lead to more quantitative estimates of ice content or even temporal changes of ice content using observations alike.

I would like to raise a few points, hoping to improve the clarity and rigor of the paper.

We are grateful to the reviewer for their helpful suggestions and comments, which we address below.

1. Excess ground ice

For TC readership, it would be better to first define what excess ground ice is, e.g., referring to the Glossary of permafrost and related ground-ice terms as 'the volume of ice in the ground which exceeds the total pore volume that the ground would have under natural unfrozen conditions'. A clear definition would help to interpret the schematics as shown in Figure 1 and equation 1. Since the excess is relative to pore volume, it is more rigorous to include the contribution of pore ice. The authors' strategy is clever as the use of late-thaw-season subsidence in a warm summer implicitly removes the contribution from melting of pore ice (which happen in all summers, manifesting in the 'early-mid-season' results shown in Figs 6 and 10). The validity of equation 1 also builds on the fact that subsidence due to the melting of pore ice is negligible in late summer.
Figure 1 is confusing at first sight. The ice content profiles imply excess ice in the active layer, even in the ice poor case (or the authors mean pore ice instead of excess ice in the active layer?). And a minor note is that the label 'heave - subsidence' unnecessarily implies heave. The figure caption is already clear.

We are not sure whether we have entirely understood the comments about the pore ice contribution. In our study area, the coring data show that massive ice, segregated ice and pore ice are abundant, but their relative contribution to the excess ice contents at scales relevant to the remote sensing observation is poorly constrained. In ice-rich permafrost, pore ice is commonly a minor contribution (we believe the referee is referring to the ~10% volume decrease upon thawing, which has the biggest impact when the entire pore space is occupied by ice). Note that in that case, the relevant fraction of the pore ice volume would be part of the excess ice. In light of our lack of ground observations, we will remain agnostic to the cryostratigraphic details of the excess ice.

We will make the following changes:

- We have added the definition from the glossary in the first paragraph of the introduction.
- To clarify Fig. 1., we have reduced the excess ice content in the active layer in the figure. We additionally added the following sentence to the caption:

Early-season subsidence reflects excess ice at the top of the active layer, which may also be present in units with ice-poor permafrost (top row), such as young floodplains.

The purpose of the heave/subsidence labels is to clarify our sign convention in which downward movement corresponds to positive values. We intend to retain it.

**2. Top of permafrost**

The classic two-layer model, active layer on top of permafrost, is adopted and well suits the nature of this work. In the discussion section, the authors did introduce the more complete four-layer structure: active layer, transient layer, intermediate layer, and permafrost. As the time scale of concern is an extremely warm summer within a fewyears (three, in this case), it is very likely that the excess subsidence is due to melting of ice-rich intermediate layer or transient layer (later of which is possibly ice rich, yet the authors claimed to be ice poor, L248), instead of the permafrost below. Then, is it justifiable to claim that excess ground ice is present at the top of permafrost (part of permafrost)? I would be more comfortable to use 'near permafrost table' or alike to allow some leeway. Quantitively, excess subsidence of 5 cm may correspond to an ice-rich layer of 5-10 cm, roughly the same order of magnitude as the thickness of transition zone (transient layer + intermediate layer).

In response to these concerns, we have strengthened the discussion of the transition zone. We continue to refer to "the top of permafrost" for simplicity, but we try to account for the definitional issues that arise when one considers longer time scales.

To this end, we have added a separate paragraph in the introduction.

> The stratigraphy of permafrost-affected soils adds complexity to the link between upper-permafrost ice content and remotely sensed late-season subsidence. To describe the cryostratigraphy in ice-rich terrain, divide the long-term permafrost into three layers. The uppermost layer, the transient layer, generally has a low to moderate excess ice content, as a result of occasional deep thaw. Disappearance of the transient layer is frequently triggered by sustained warming or disturbance. The subjacent ice-rich intermediate layer is then exposed, increasing the susceptibility to enhanced subsidence. The risk of sustained thaw consolidation is amplified where the intermediate layer overlies massive ice such as ice wedges. Once the protection afforded by the transient and intermediate layer has been lost, further thaw will lead to ice wedge degradation. Ice wedge polygons also illustrate the large lateral variability in ground ice conditions, which need to be considered when interpreting late-season subsidence as an indicator of ice-rich upper permafrost.

We now state that the transient layer has low to moderate excess ice contents.

We also made changes to the discussion section. Most importantly, we strengthened discussion of how we suspect the transient layer contributed to the difference between 2018 and 2019:

> Inter-annual variability in late-season subsidence of ice-rich areas poses challenges for ground ice mapping. Potential sources of inter-annual variability and trends include surface changes (e.g., soil moisture, disturbance, snow) and variable meteorological conditions such as precipitation. Memory effects could also be relevant. Taking the ice-rich area in Figure 8a) as an example, we speculate that thaw of materials with moderate excess ice contents (transient layer) at the end of the warm summer of 2018 (limited late-season subsidence of ~ 2 cm) could have promoted larger subsidence in 2019 by weakening the protection given to the subjacent materials richer in excess ice (intermediate layer, massive ice). Equally, the summer of 2018 may not have been warm (and wet) enough to allow for reliable identification of the vulnerable ground ice at this location. That the identification strategy presupposes an initial degradation of ground ice constitutes its biggest limitation.

**3. Extremely warm summer**

I have no doubt that 2019 was an extremely warm summer in Kivalina (it was the warmest according to Fig. 2b). Without a statistical or meteorological perspective, I would also regard 2018 as very warm (2nd warmest in the Fig 2b time series); yet the late-summer subsidence in 2018 was normal. Then I was wondering how 'exceptional' the warming must be to cause the excess subsidence. Is it ever possible that the excess subsidence in 2019 resulted from the decadal warming in the region, esp. considering that it typically takes decades or longer to thaw permafrost in continuous permafrost zones?

We agree that the inter-annual variability raises important questions. In addition to temperature, we mention complicating factors such as winter conditions (e.g., snow) and soil moisture. We agree that legacy effects such as the general warming trend and the warm preceding summer of 2018 are relevant. We have added a specific example:

> Inter-annual variability in late-season subsidence of ice-rich areas poses challenges for ground ice mapping. Potential sources of inter-annual variability and trends include surface changes (e.g., soil moisture, disturbance, snow) and variable meteorological conditions such as precipitation. Memory effects could also be relevant. Taking the ice-rich area in Figure 8a) as an example, we speculate that thaw of materials with moderate excess ice contents (transient layer) at the end of the warm summer of 2018 (limited late-season subsidence of ~ 2 cm) could have promoted larger subsidence in 2019 by weakening the protection given to the subjacent materials richer in excess ice (intermediate layer, massive ice). Equally, the summer of 2018 may not have been warm (and wet) enough to allow for reliable identification of the vulnerable ground ice at this location. That the identification strategy presupposes an initial degradation of ground ice constitutes its biggest limitation.

In addition, we will include precipitation data (despite concerns about gaps)

4. Timings of late thaw season

The choices of the beginning and end of late season (namely, t1 and t2) make sense. I was just wondering if these are backed up by temperature data and do you need to adjust them when applying this method in different areas (I suppose you need). The authored mentioned 'diurnal frost heave', were there any signs of heave in September? Then how about late-summer heave (Mackay 1983 10.1139/e83-012)?

We agree that these choices are of practical relevance. We thus plan to add sensitivity analyses, i.e. we plan to see how the results in Fig. 8 change when the beginning and duration are altered.

Late-season frost heave is a particular concern, especially in areas or years with freezing in late August/early September. We did not notice any clear signs of frost heave in this data set, but they are well documented in the literature.

We agree that we did not give sufficient attention to summer heave. We have added that we "neglect summer heave due to water movement into frozen materials", and we cite R. Mackay's classic paper.

5. Overall, I think larger-than-usual late-season subsidence is an indirect, instead of direct, indicator of "vulnerable excess ground ice" near the permafrost table.

We have decided not to refer to late-season subsidence as a direct indicator in the abstract any more. Arguably, directness is a matter of degree rather than of kind.

Minor comments

3.1.4 The authors may provide more details of the spline fitting and help us to understand

Fig 3b.

L167: $ should be &

L209: Delete the extra for

L228: indicate should be indicates

L251: Its should be It

L312: an should be and

L316: includes should be include

Thank you. We have fixed these errors. We have expanded the description of the spline fitting process, now describing how we used ordinary least squares to estimate the coefficients of the spline expansion.

---

## Author Comment (AC4) · 19 Jan 2021

We would like to express our gratitude to the three referees for their helpful comments and suggestions, which we believe will help us improve the quality clarity of our manuscript.

We now summarize the key changes we intend to make.

1) We will add new figures / panels.

a) We will also show precipitation observations (despite the large number of missing values).

[Figure]

b) We will study the sensitivity of the late-season subsidence observations to the specification of the late-season period. The associated figure(s) will likely be added to the supplement.

c) We intend to add a panel to Fig. 7 that shows classification results based on thresholding the late-season subsidence. This will make it easier to compare to the independent ground ice map.

2) We will make numerous minor to moderate edits in the text. The most important ones serve to clarify aspects concerning

a) cryostratigraphy and interpretation of late-season subsidence;

b) spatial variability;

c) open questions and future developments.

We provide detailed point-by-point answers in separate documents. There, we describe how we intend to modify the manuscript. Most of the minor edits have already been implemented in an internal revised version of the manuscript because we find it is easier to keep track of them this way.

––––––––––––––––––––––––––––––

---

## Author Response (AR2)

Dear Peter,

Thank you for your additional comments. They have been instrumental in improving the presentation of our results. In our response, we focus on the three major points you identified in your comments.

**1) Subsidence time series**

We have changed all the subsidence time series figures. Subsidence is plotted downward, and the zero point is at the beginning of the time series. Figure 1 now looks like this:

[Figure]

We now also clarify that the subsidence is cumulative. The instantaneous subsidence is consistently referred to as subsidence rate.

A side effect of these changes is that it is more difficult for the reader to deduce the late-season subsidence component. It is particularly difficult to make inter-annual comparisons. This necessitated additional explanations in Fig. 1, the captions and the text.

**2) Focus**

We have restructured the abstract and the introduction to better emphasize our central contribution, which is to assess the suitability of remotely-sensed late season subsidence as an indicator of ice-rich materials in the upper permafrost.

We have strengthened the literature review and now explicitly identify the knowledge gap that the "the suitability of these observations for identifying ice-rich permafrost is unknown." We address how our study differs from the InSAR literature.

The conclusions have been re-arranged and tightened accordingly.

**3) Cryostratigraphy**

We now devote less attention to the cryostratigraphic classification of Shur et al., but we have not eliminated reference to it altogether. The paragraph in the introduction, which we added at the referees' behest, has been removed. The cryostratigraphic interpretation of the Sentinel-1 observations in the discussion has been greatly reduced. While we would rather not interpret the Sentinel-1 observations, the insightful referee reports indicated that many readers will appreciate an explicit link to published field observations.

The challenge of finding a universally accepted terminology is formidable. The terms intermediate, transition, and transient layer have become entrenched, at least in Alaska. But we are aware that the usage is somewhat inconsistent. On a related note, we would prefer to avoid using the expressions active layer and permafrost on multi-annual time scales, but these expressions have also become entrenched.

**Minor changes**

- Removed titles from the figures and updated captions
- Flipped the vertical colour bars in the subsidence maps.
- We do not talk about vulnerable ground ice anymore; the title has been changed accordingly.
- Added quantitative statistical information (mean, standard deviation) about the inter-annual difference in late-season subsidence to the results.
- Aspect: Added a description that contrasts east and west-facing slopes
- Streamlined the description of the manual mapping exercise

These changes are highlighted in a separate version of the manuscript that we have include in the submission.

 Sincerely,

Simon Zwieback

---

## Editor Decision (ED2)

[revised manuscript text omitted]

|---|---|---|---|
| 05-1 | 67.786/-164.474 | 1 | perimeter of low-centred polygon in uplands
massive ice |
| 05-2 | 67.786/-164.474 | 1 | interior of low-centred polygon in uplands
silt with up to 40% visible ice |
| 05-3 | 67.811/-164.563 | 1 | tussock-covered hillslope
silt with 40% visible ice |
| 05-4 | 67.816/-164.598 | 1 | non-sorted circle in polygon interior; gentle hillslope
massive ice |
| 05-5 | 67.821/-164.592 | 1 | tussock-covered interior of; gentle hillslope
massive ice and ice-rich silt (30% visible ice) |
| 05-6 | 67.823/-164.609 | 1 | interior of low-centred polygon on edge of drainage
ice inclusions in peat and massive ice |
| 05-7 | 67.845/-164.735 | 0 | gravel-covered bench
gravel with 5% visible ice |
| 05-8 | 67.849/-164.726 | 1 | tussock-covered interior of faint polygon
colluvial silt with 30 to 40% visible ice |
| 05-9 | 67.847/-164.731 | 1 | tussock-covered colluvium
silt and gravel: 50% visible ice |
| 05-10 | 67.843/-164.740 | 1 | tussock-covered interior of faint polygon
colluvial silt 30% visible ice (lenses up to 5 mm thick) |
| 05-11 | 67.845/-164.725 | 1 | tussock-covered hillslope; colluvium
ice with soil inclusions: 50 to 60% visible ice |
| 05-12 | 67.847/-164.721 | 1 | tussock-covered hillslope; colluvium
sandy, gravelly silt with 40 to 50% visible ice |
| 05-13 | 67.851/-164.732 | 1 | tussock-covered hillslope; colluvium
sandy silt with ice lenses up to 5 mm thick |
| 05-14 | 67.849/-164.737 | 1 | tussock-covered hillslope; colluvium
silt with 15–20% visible ice; ice with silt inclusions |

[Figure]

Figure S1: Meteorological time series for Kivalina from the MERRA-2 reanalysis: a) air temperature $T_a$; b) monthly precipitation for 2017–2019 and the 30-year average. Large precipitation amounts exceeding 100 mm in July 2019 are confirmed by in-situ observations, whose coverage during the rest of the study period is poor.

[Figure]

Figure S2: Early–mid-season subsidence (10 June – 10 August) for 2017 and 2019; otherwise identical to Fig. ??a–b)

[Figure]

**Figure S3:** Contour plots of a kernel density estimate of the early–mid-season and late-season subsidence for ice rich (purple) and ice poor (grey), as determined independently by manual mapping. The columns correspond to the year; the rows to the beginning of the late season: baseline: 10 August, early start: 31 July, late start: 20 August.

---

## Author Response (AR3)

Dear Peter,

Thank you for your detailed comments and suggestions. They have been instrumental in further improving the clarity of the manuscript.

The only suggestion that we only partially incorporated relates to rapid thaw. We have extended the paragraph in the introduction to stress i) that late-season subsidence can constitute a significant geomorphic change by itself, and ii) its potential as a precursor for long-term instability. We also mention its complementarity to field observations. We do not address the potential link to hillslope failures such as active layer detachments because we lack detailed observations. This is an exciting avenue for future research. On the Seward Peninsula, colleagues have observed hundreds of microfailures on hillslopes whose movement InSAR indicates accelerated substantially at the end of the record summer of 2019. Unfortunately, they could not revisit the site in 2020.

The following list presents a summary of our changes:

- We now contrast our approach, which focuses on a single summer, with previous long-term InSAR studies. We also stress that late-season subsidence in a single year can constitute a geomorphologically significant change.
- We emphasize the role of precipitation and moisture, as well as the fact that 2019 was very wet more strongly. We have added relevant references, also citing additional papers that focus on surface disturbance as a trigger of thermokarst for balance.
- We now devote more space (results section, figure caption) to negative late-season subsidence observations. In addition to clarifying that they correspond to heave between 10 August and 10 September, we also explicitly state that their magnitude is comparable to the observational accuracy. We have retained the kernel density plots because we contend they accurately reflect that there are negative estimates and because the number of data points is so large that scatter plots would be difficult to comprehend.
- We improved Fig. 3, most notably by labelling the three basis functions.
- We have adjusted the colourbar of the late-season subsidence maps so that they correspond to the spread of the data (no large negative values).
- We have fixed the erroneous figure references in Fig. 7 and the supplement.
- We have tried to end each paragraph in the subsection on limitations on a high note, by giving constructive suggestions on how these limitations can be mitigated.

Please find the revised manuscript and supplement as well as a version with tracked changes attached.

Thank you for your support and suggestions.

Kind regards,

Simon Zwieback